# An essential, kinetoplastid-specific GDP-Fuc: β-D-Gal α-1,2-fucosyltransferase is located in the mitochondrion of *Trypanosoma brucei*

Giulia Bandini[1†], Sebastian Damerow[1†], Maria Lucia Sempaio Guther[1], Hongjie Guo[2], Angela Mehlert[1], Jose Carlos Paredes Franco[1], Stephen Beverley[2], Michael AJ Ferguson[1]*

[1]Wellcome Centre for Anti-Infectives Research, School of Life Sciences, University of Dundee, Dundee, United Kingdom; [2]Department of Molecular Microbiology, Washington University School of Medicine, St. Louis, United States

**Abstract** Fucose is a common component of eukaryotic cell-surface glycoconjugates, generally added by Golgi-resident fucosyltransferases. Whereas fucosylated glycoconjugates are rare in kinetoplastids, the biosynthesis of the nucleotide sugar GDP-Fuc has been shown to be essential in *Trypanosoma brucei*. Here we show that the single identifiable *T. brucei* fucosyltransferase (TbFUT1) is a GDP-Fuc: β-D-galactose α-1,2-fucosyltransferase with an apparent preference for a Galβ1,3GlcNAcβ1-O-R acceptor motif. Conditional null mutants of *TbFUT1* demonstrated that it is essential for both the mammalian-infective bloodstream form and the insect vector-dwelling procyclic form. Unexpectedly, TbFUT1 was localized in the mitochondrion of *T. brucei* and found to be required for mitochondrial function in bloodstream form trypanosomes. Finally, the *TbFUT1* gene was able to complement a *Leishmania major* mutant lacking the homologous fucosyltransferase gene (Guo et al., 2021). Together these results suggest that kinetoplastids possess an unusual, conserved and essential mitochondrial fucosyltransferase activity that may have therapeutic potential across trypanosomatids.

**\*For correspondence:**
m.a.j.ferguson@dundee.ac.uk

†These authors contributed equally to this work

**Competing interest:** The authors declare that no competing interests exist.

## Introduction

The protozoan parasites of the *Trypanosoma brucei* group are the causative agents of human and animal African trypanosomiasis. Bloodstream form *T. brucei* is ingested by the tsetse fly vector and differentiates into procyclic form parasites to colonize the tsetse midgut. To then infect a new mammalian host, *T. brucei* undergoes a series of differentiations that allows it to colonize the fly salivary gland and to be transferred to a new host during a subsequent blood meal (*Matthews, 2005*).

The surface coat of the bloodstream form is characterized by the GPI-anchored, *N*-glycosylated and occasionally *O*-glycosylated variant surface glycoprotein (VSG) (*Cross, 1996*; *Mehlert et al., 1998*; *Pays and Nolan, 1998*; *Pinger et al., 2018*; *Schwede and Carrington, 2010*), while procyclic cells express a family of GPI-anchored proteins called procyclins (*Richardson et al., 1988*; *Roditi et al., 1998*; *Treumann et al., 1997*; *Vassella et al., 2001*), free glycoinositolphospholipids (*Nagamune et al., 2004*; *Roper et al., 2005*; *Vassella et al., 2003*), and a high molecular weight glycoconjugate complex (*Güther et al., 2009*). The importance of glycoproteins to parasite survival and infectivity has led to the investigation of enzymes for GPI anchor biosynthesis (*Chang et al., 2002*; *Nagamune et al., 2000*; *Smith et al., 2004*; *Urbaniak et al., 2014*; *Urbaniak et al., 2008*) and nucleotide sugar biosynthesis (*Bandini et al., 2012*; *Denton et al., 2010*; *Kuettel et al.,*

*2012*; *Sampaio Guther et al., 2021*; *Marino et al., 2010*; *Marino et al., 2011*; *Roper et al., 2002*; *Roper et al., 2005*; *Stokes et al., 2008*; *Shaw et al., 2003*; *Turnock et al., 2007*; *Urbaniak et al., 2006a*; *Urbaniak et al., 2006b*; *Urbaniak et al., 2013*; *Zmuda et al., 2019*) as potential therapeutic targets.

Nucleotide sugars are used as glycosyl donors in many glycosylation reactions. GDP-fucose (GDP-Fuc) was identified in the nucleotide sugar pools of *T. brucei*, *Trypanosoma cruzi,* and *Leishmania major* (*Turnock and Ferguson, 2007*), and its biosynthesis is essential for parasite growth in procyclic and bloodstream form *T. brucei* (*Turnock et al., 2007*) and in *L. major* promastigotes (*Guo et al., 2017*). Interestingly, *T. brucei* and *L. major* use different pathways to synthesize GDP-Fuc. *T. brucei* utilizes the de novo pathway in which GDP-Fuc is synthesized from GDP-mannose via GDP-mannose-4,6-dehydratase (GMD) and GDP-4-keto-6-deoxy-D-mannose epimerase/reductase (GMER) (*Sampaio Guther et al., 2021*; *Turnock et al., 2007*; *Turnock et al., 2007*). Conversely, *L. major* has two related bifunctional D-arabinose/L-fucose kinase/pyrophosphorylase, AFKP80 and FKP40, that synthesize GDP-Fuc from free fucose (*Guo et al., 2017*). Despite the aforementioned essentialities for GDP-Fuc in *T. brucei* and *L. major*, the only structurally defined fucose-containing oligosaccharide in trypanosomatids is the low-abundance Ser/Thr-phosphodiester-linked glycan on *T. cruzi* gp72, a glycoprotein that has been implicated in flagellar attachment (*Allen et al., 2013*; *Cooper et al., 1993*; *Ferguson et al., 1983*; *Haynes et al., 1996*).

Fucosyltransferases (FUTs) catalyse the transfer of fucose from GDP-Fuc to glycan and protein acceptors and are classified into two superfamilies (*Coutinho et al., 2003*; *Lombard et al., 2014*). One superfamily contains all α1,3/α1,4-FUTs (carbohydrate active enzyme, CAZy, family GT10) and the other contains all α1,2-, α1,6-, and protein *O*-fucosyltransferases (GT11, GT23, GT37, GT56, GT65, GT68, and GT74; *Martinez-Duncker, 2003*). In eukaryotes, FUTs are generally either type II transmembrane Golgi proteins or ER-resident enzymes (POFUT1 and POFUT2) (*Breton et al., 1998*), but two exceptions have been described: (i) PgtA, a cytoplasmic bifunctional β1,3-galactosyltransferase α1,2-FUT found in *Dictyostelium discoideum* and *Toxoplasma gondii* (*Rahman et al., 2016*; *Van Der Wel et al., 2002*) that is part of an oxygen-sensitive glycosylation pathway that attaches a pentasaccharide to the Skp1-containing ubiquitin ligase complex (*West et al., 2010*); and (ii) SPINDLY, a protein *O*-fucosyltransferase that modifies nuclear proteins in *Arabidopsis thaliana* and *T. gondii* (*Gas-Pascual et al., 2019*; *Zentella et al., 2017*).

*T. brucei* and other kinetoplastids contain a single mitochondrion. In the bloodstream form of the parasite, this organelle has a tubular structure, while in the procyclic form it is organized in a complex network with numerous cristae, reflecting the absence and presence, respectively, of oxidative phosphorylation (*Matthews, 2005*; *Priest and Hajduk, 1994*). The parasite mitochondrion is further characterized by a disc-shaped DNA network called the kinetoplast (*Jensen and Englund, 2012*) that is physically linked with the flagellum basal body (*Ogbadoyi et al., 2003*; *Povelones, 2014*).

While secretory pathway and nuclear/cytosolic glycosylation systems have been studied extensively, little is known about glycosylation within mitochondria. A glycoproteomic approach in yeast revealed several mitochondrial glycoproteins (*Kung et al., 2009*), but it was not determined whether these were imported from the secretory pathway or glycosylated within the mitochondria by as yet unknown glycosyltransferases. The only characterized example of a mitochondrial glycosyltransferase is the mitochondrial isoform of mammalian *O*-GlcNAc transferase (OGT). *O*-GlcNAcylation is a cycling modification, involved in signalling, in which OGT adds GlcNAc to Ser/Thr residues and *O*-GlcNAcase (OGA) removes it (*Bond and Hanover, 2015*). Studies have shown that both mitochondrial OGT (mOGT) and OGA are present and active in mammalian mitochondria and putative mitochondrial targets have been identified (*Banerjee et al., 2015*; *Sacoman et al., 2017*). Further, a mammalian mitochondrial UDP-GlcNAc transporter associated with mitochondrial *O*-GlcNAcylation has been described (*Banerjee et al., 2015*). However, orthologues of OGT and OGA genes are not present in kinetoplastids.

Here, we report on a gene (*TbFUT1*) encoding a mitochondrial α-1,2-fucosyltransferase protein (TbFUT1) in *T. brucei* that is essential to parasite survival. Similar results were obtained in the related trypanosomatid parasite *L. major* (*Guo et al., 2021*), extending this unexpected finding across the trypanosomatid protozoans.

## Results

### Identification, cloning, and sequence analysis of TbFUT1

The CAZy database lists eight distinct FUT families (see Introduction) (*Lombard et al., 2014*). One or more sequences from each family were selected for BLASTp searches of the predicted proteins from the *T. brucei, T. cruzi,* and *L. major* genomes (*Supplementary file 1*). Strikingly, only one putative fucosyltransferase gene (*TbFUT1*) was identified in the *T. brucei* genome (GeneDB ID: Tb927.9.3600) belonging to the GT11 family, which is comprised almost exclusively of α-1,2-FUTs (*Coutinho et al., 2003*; *Zhang et al., 2010*). Homologues of *TbFUT1* were also found in the *T. cruzi* and *L. major* genomes and, unlike *T. brucei, T. cruzi,* and *L. major,* also encode for GT10 FUT genes (*Supplementary file 1*). Finally, *Leishmania* spp. express a family of α-1,2-arabinopyranosyltransferases (SCA1/2/L, CAZy family GT79) that decorate phosphoglycan side chains and have been suggested to act as FUTs in presence of excess fucose (*Guo et al., 2021*).

The TbFUT1 predicted amino acid sequence shows relatively low sequence identity to previously characterized GT11 FUTs, for example, *Helicobacter pylori* (26%) or human FUT2 (21%) (*Kelly et al., 1995*; *Wang et al., 1999*). Nevertheless, conserved motifs characteristic of this family can be identified (*Figure 1A*; *Li et al., 2008*; *Martinez-Duncker, 2003*). Motif I (aa 153–159) is shared with α-1,6-FUTs and has been implicated in the binding of GDP-Fuc (*Takahashi et al., 2000*), whereas no clear functions have yet been assigned to motifs II, III, and IV (aa 197–207, 265–273, and 13–18, respectively). Notably, TbFUT1 lacks an identifiable *N*-terminal signal peptide and has only a low-confidence prediction for a type II membrane protein *N*-terminal transmembrane domain (0.34108 on TMHMM-2.0 for residues 6–28). These two features would be expected of a typical Golgi-localized FUT (*Breton et al., 1998*; *Figure 1A*). Indeed, further analysis of the TbFUT1 predicted amino acid sequence using *PSort II* (43.5 % mitochondrion, 4.3 % secretory pathway, 26.1 % cytosolic, *Horton and Nakai, 1997*), *Target P* (0.428 mitochondrial, 0.064 secretory, 0.508 other, *Emanuelsson et al., 2000*), and *Mitoprot* (0.5774 probability of mitochondrial localization, *Claros and Vincens, 1996*) rated the localization as most likely mitochondrial and identified a putative mitochondrial targeting motif (M/L … RR) with RR at sequence positions 30 and 31. Conservation of this eukaryotic targeting motif has been previously shown for other parasite mitochondrial proteins (*Krnáčová et al., 2012*; *Long et al., 2008*).

Additional BLASTp searches showed that there is generally a single TbFUT1 gene homologue in each kinetoplastid species, and a phylogram of FUT sequences indicates that TbFUT1 homologues form a distinct clade closest to bacterial α-1,2-FUT (*Figure 1B*).

### Recombinant expression of TbFUT1

The *TbFUT1* ORF was amplified from *T. brucei* 427 genomic DNA and cloned in the pGEX6P1 expression vector. The resulting construct (pGEX6P1-GST-PP-*TbFUT1*) encoded for the *TbFUT1* ORF with a glutathione-S-transferase (GST) tag at its N-terminus and a PreScission Protease (PP) cleavage site between the two protein-encoding sequences. Sequencing confirmed what was subsequently deposited at TriTrypDB (Tb427_090021700) and identified two amino acid differences between TbFUT1 in the 927 and 427 strains (L185V and T232A).

The pGEX6P1-GST-PP-*TbFUT1* construct was expressed in *Escherichia coli* and the fusion protein purified as described in Materials and methods. The identities of the two higher molecular weight bands (*Figure 2—figure supplement 1*, lane 8) were determined by peptide mass fingerprinting. The most abundant band was identified as TbFUT1, while the fainter band was identified as a subunit of the *E. coli* GroEL chaperonin complex. The apparent molecular weight of GST-PP-TbFUT1 chimeric protein (57 kDa) was consistent with the predicted theoretical molecular weight (58.1 kDa).

### Recombinant TbFUT1 is active in vitro

The activity of recombinantly expressed GST-TbFUT1 fusion protein was tested by incubation with GDP-[$^3$H]Fuc, as a donor, and a panel of commercially available mono- to octasaccharides (*Table 1*) selected from the literature as possible α-1,2-FUT substrates (*Li et al., 2008*; *Wang et al., 1999*; *Zhang et al., 2010*). The effectiveness of each acceptor was evaluated based on the presence/absence and intensities of the TLC bands corresponding to the radiolabelled reaction products (*Figure 2* and *Table 1*). GST-TbFUT1 showed best activity with Galβ1,3GlcNAc (LNB) (*Figure 2*, lane 2) and its β-O-methyl glycoside (*Figure 2*, lane 21). Other larger oligosaccharides containing Galβ1,3GlcNAcβ1-O-R as a terminal motif (LNT and LNH) were also good acceptors (*Figure 2*, lanes 11 and 15), with the

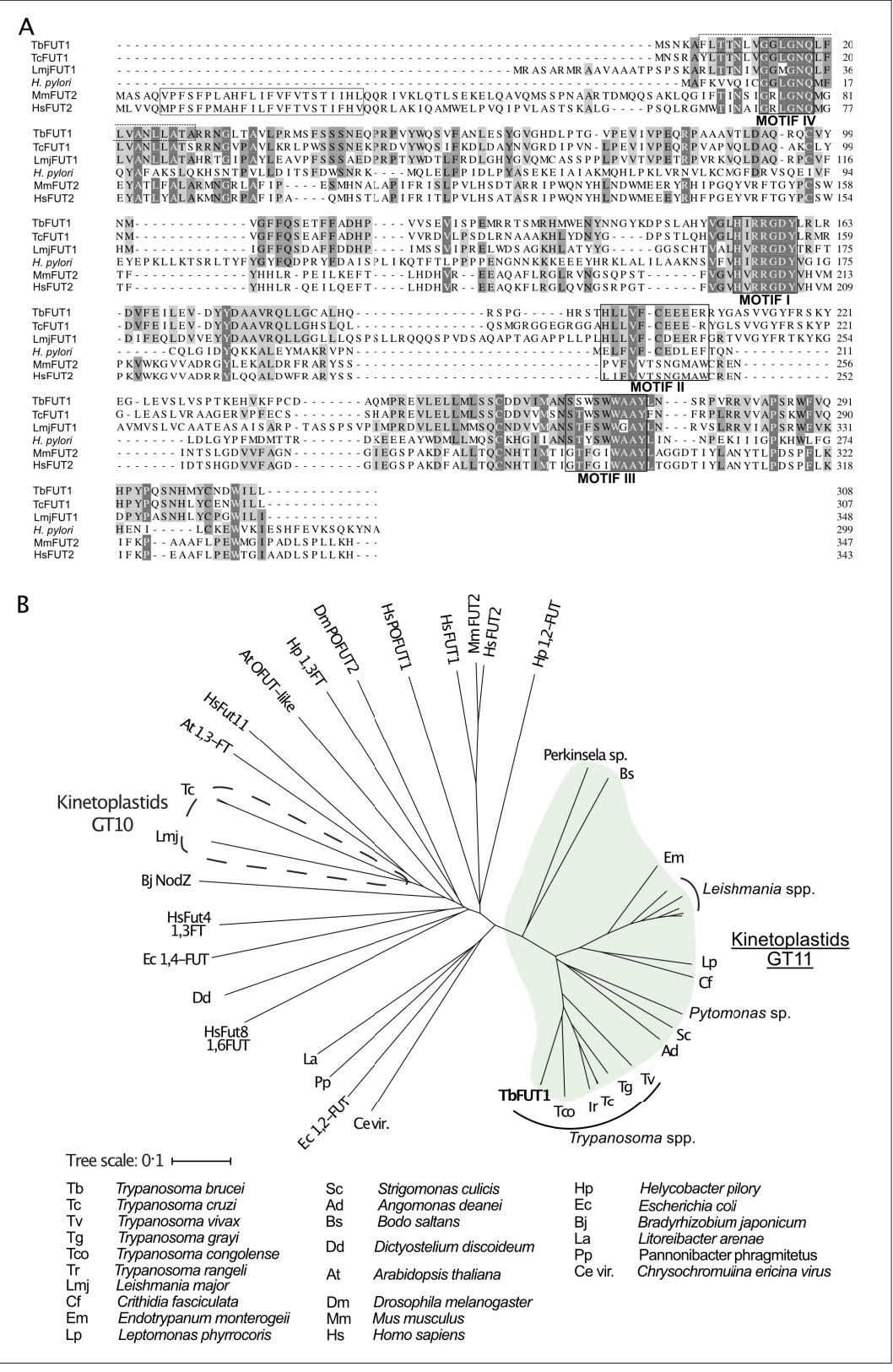

**Figure 1.** Amino acid sequence and phylogenetic analyses of TbFUT1. (**A**) Sequence alignment of TbFUT1 and other GT11 family fucosyltransferases (FUTs) shows that TbFUT1 (Tb927.9.3600) lacks a conventional *N*-terminal type 2 membrane protein transmembrane domain (TM, grey box) but contains conserved motifs I–IV (*black boxes*). A putative TbFUT1 TM (*dashed box*) overlaps with motif IV but does not align with the eukaryotic FUT TM and is

*Figure 1 continued on next page*

*Figure 1 continued*

most likely part of a cleavable *N*-terminal mitochondrial targeting sequence (residues 1–31 with cleavage at R30–R31). Sequences used in the alignment: *T. cruzi* (TcCLB.506893.90), *L. major* (LmjF01.0100), *H. pylori* (AAC99764), *Homo sapiens* FUT2 (AAC24453), and *Mus musculus* FUT2 (AAF45146). (**B**) A selection of known and predicted fucosyltransferase protein sequences was aligned using ClustalΩ (***Sievers et al., 2011***), and the unrooted phylogram shown was generated by iTOL (itol.embl.de) (***Letunic and Bork, 2016***). Single homologues of the GT11 TbFUT1 were found in all the kinetoplastids (marked in *green*) and, collectively, these sequences form a clade distant from other fucosyltransferase sequences, including the other kinetoplastid GT10 fucosyltransferases (marked by a *dashed line*). No TbFUT1 homologues were found in apicomplexan parasites or in *Euglena*.

exception of iLNO (***Figure 2***, lane 13). Lactose was also recognized (***Figure 2***, lane 1), while LacNAc and the LacNAc-terminating branched hexasaccharide LNnH were weak acceptors (***Figure 2***, lanes 3 and 10). Interestingly, TbFUT1 was also able to transfer fucose to 3′-fucosyllactose, albeit inefficiently (***Figure 2***, lane 16), whereas no transfer could be seen to Galβ1,6GlcNAc (***Figure 2***, lane 17) or to free Gal or β-Gal-O-methyl (***Figure 2***, lanes 9 and 20). As expected, no products were observed when acceptor oligosaccharides were omitted from the reaction (***Figure 2***, lane 4). To confirm the detected activities were specific to the recombinant GST-TbFUT1, and not due to some co-purifying endogenous *E. coli* contaminant, the assay was also performed using material prepared from *E. coli* expressing the empty pGEX6P1 vector. No transfer of radiolabelled fucose could be observed under these conditions (***Figure 2***, lanes 5–7).

A band with the same mobility as free fucose was observed in all assay reactions and was considerably stronger in the presence of the GST-TbFUT1 preparation (***Figure 2***, lanes 1–4 and 9–22) than when GDP-[$^3$H]Fuc was incubated with reaction buffer alone (***Figure 2***, lane 8) or in the presence of

**Table 1.** Acceptor substrates and semi-quantitative fucosyltransferase activities.

| TbFUT1 activity | Lane of Figure 1 | Abbreviations* | Name | Structure |
|---|---|---|---|---|
| +++ | 2, 6 | LNB | Lacto-N-biose | Galβ1,3GlcNAc |
| +++ | 21 | LNB-OMe | Lacto-N-biose-O-methyl | Galβ1,3GlcNAcβ-OMe |
| ++ | 11 | LNT | Lacto-N-tetraose | Galβ1,3GlcNAcβ1,3Galβ1,4Glc |
| ++ | 14 | LNH | Lacto-N-hexaose | Galβ1,3GlcNAcβ1,3(Galβ1,4GlacNAcβ1,6)Galβ1,4Glc |
| ++ | 1,5 | Lac | Lactose | Galβ1,4Glc |
| + | 13 | iLNO | Iso-lacto-N-octaose | Galβ1,3GlcNAcβ1,3(Galβ1,3GlcNAcβ1,3Galβ1,4GlcNAcβ1,6)Galβ1,4Glc |
| + | 3,7 | LacNAc | N-acetyllactosamine | Galβ1,4GlcNAcN |
| + | 10 | LNnH | Lacto-N-neohexaose | Galβ1,4GlcNAcβ1,3(Galβ1,4GlcNAcβ1,6)Galβ1,4Glc |
| + | 12 | LNnT | Lacto-N-neotetraose | Galβ1,4GlcNAcβ1,3Galβ1,4Glc |
| + | 16 | 3′-FL | 3′-Fucosyllactose | Galβ1,4(Fucα1,3)Glc |
| – | 9 | β-Gal | β-Galactose | β-Gal |
| – | 15 | GNG | β1,6-Galactosyl-N-acetyl-glucosamine | Galβ1,6GlcNAc |
| – | 17 | 1,6 GB | β1,6-Galactobiose | Galβ1,6Gal |
| – | 18 | 1,4 GB | Galabiose | Galα1,4Gal |
| – | 19 | LB2TS | Linear B2 trisaccharide | Galα1,3Galβ1,4GlcNAc |
| – | 20 | β-Gal-OMe | β-Galactose-O-methyl | β-Gal-OMe |
| – | 22 | 2′-FL | 2′-Fucosyllactose | Fucα1,2Galβ1,4Glc |

*The relative efficiencies of the acceptors as TbFUT1 substrates (+++, ++, +, and –) are based on visual inspection of the intensities of the products bands in Figure 2.

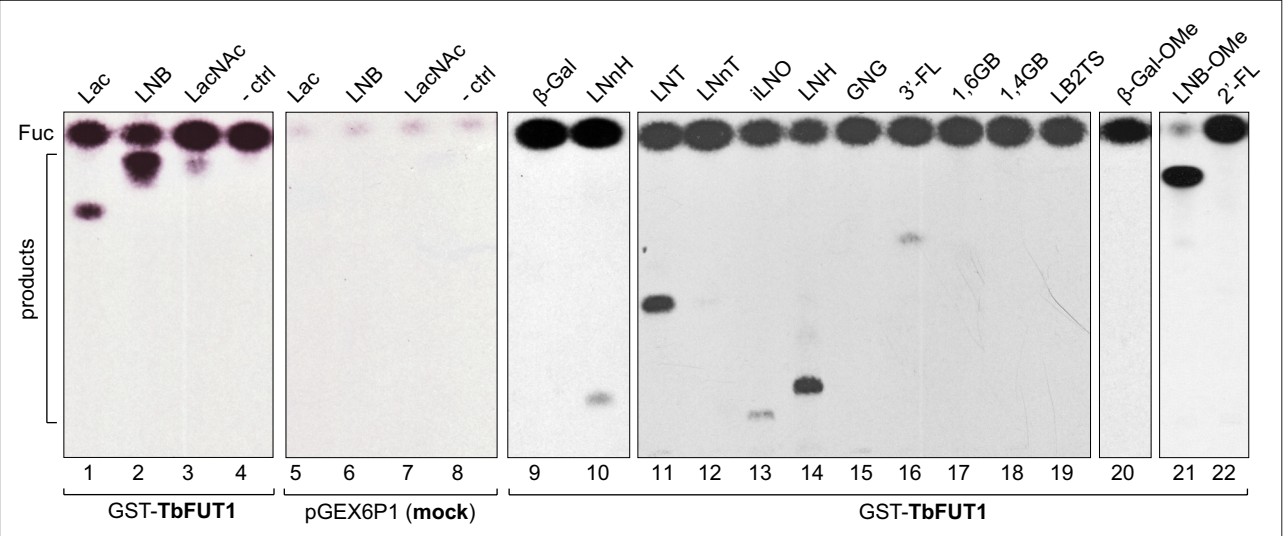

**Figure 2.** Recombinant GST-TbFUT1 transfers [³H]Fuc to a variety of sugar acceptors. Each assay used 2 µg of purified GST-TbFUT1, GDP-[³H]Fuc, and 1 mM of acceptor. Reaction products were desalted and separated by silica High Performance Thin Layer Chromatography (HPTLC) and detected by fluorography. LNB, LNB-OMe, and LNB-terminating structures were the best acceptors tested. The acceptor abbreviations above each lane are defined in Table 1. -ctrl: negative control reaction without acceptor (lane 4) or with buffer alone (lane 8).

The online version of this article includes the following figure supplement(s) for figure 2:

**Figure supplement 1.** Purification of recombinant GST-TbFUT1.

**Figure supplement 2.** TbFUT1 activity is independent of divalent metal cations.

the material purified from the *E. coli* cells transformed with the empty vector (*Figure 2*, lanes 5–7). These data suggest that TbFUT1 has a significant propensity to transfer Fuc to water. Interestingly, one of the substrates (LNB-O-Me; Galβ1,3GlcNAcβ1-O-methyl) suppressed the amount of free Fuc produced in the reaction (*Figure 2*, lane 21), suggesting that this glycan may bind more tightly to the TbFUT1 acceptor site than the other oligosaccharides tested and thus prevent the transfer of Fuc from GDP-Fuc to water. In vitro high sugar nucleotide hydrolysis activity has been previously described for at least one member of the GT11 family (*Zhang et al., 2010*).

Inverting α-1,2 and α-1,6-FUTs do not usually require divalent cations for their activity (*Beyer and Hill, 1980*; *Kamińska et al., 2003*; *Li et al., 2008*; *Pettit et al., 2010*). To study the divalent cation dependence of TbFUT1, the assay was repeated in buffer without divalent cations or containing EDTA. No change in activity was observed in either case, indicating TbFUT1 does not require divalent cations for its activity (*Figure 2—figure supplement 2*).

Finally, given the propensity for TbFUT1 to transfer [³H]Fuc from GDP-[³H]Fuc to water, producing free [³H]Fuc and presumably GDP, we set up a GDP-Glo assay similar to that used for LmjFUT1 (*Guo et al., 2021*) to monitor the turnover by recombinant TbFUT1 of non-radioactive GDP-Fuc to GDP (see Materials and methods). In this assay, we could see TbFUT1-dependent turnover of GDP-Fuc to GDP in the absence of acceptor substrate (70 ± 2 pmol) and the stimulation of turnover in the presence of LNB acceptor substrate (265 ± 13 pmol), consistent with the results in the radiometric assay. Under the same conditions, there was no detectable turnover of either GDP-Man or GDP-Glc, showing that TbFUT1 is specific, or at least highly selective, for GPD-Fuc as donor substrate.

## Characterization of the TbFUT1 reaction product

The glycan reaction products were structurally characterized to determine the anomeric and stereo-chemical specificity of TbFUT1. Initially, we performed exoglycosidase and/or acid treatment of the radiolabelled reaction products (recovered by preparative TLC) utilizing Lac, LacNAc, and LNB as substrates. The tritium label ran with the same mobility as authentic Fuc after acid hydrolysis of all three products (*Figure 3—figure supplement 1A,C*) and after *Xanthomonas manihotis* α-1,2-fucosidase digestion of the Lac and LNB products (*Figure 3—figure supplement 1B,C*). These data suggest that [³H]Fuc was transferred in α1,2 linkage to the acceptor disaccharides.

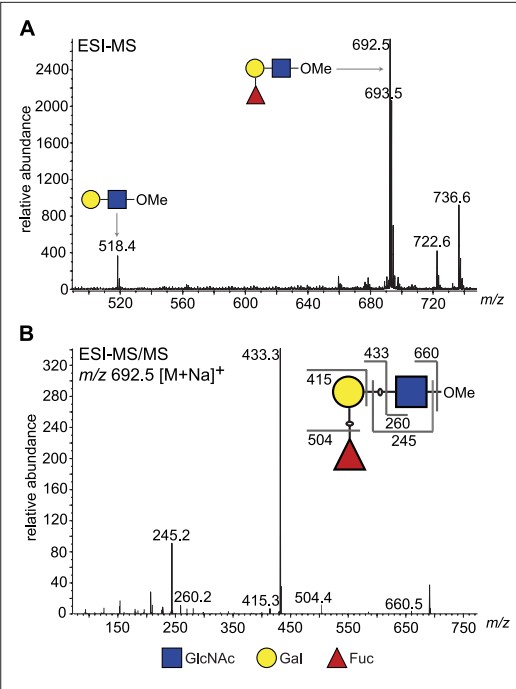

**Figure 3.** ESI-MS and ESI-MS/MS of TbFUT1 in vitro reaction product. (**A**) ESI-MS of the purified and permethylated reaction product. The ion at $m/z$ 692.5 is consistent with the $[M + Na]^+$ ion of a permethylated trisaccharide of composition $dHex_1Hex_1HexNAc_1$. Some of the unmodified acceptor ($Hex_1HexNAc_1$) was also observed ($m/z$ 518.4). (**B**) MS/MS product ion spectrum of $m/z$ 692.5. The collision-induced fragmentation pattern indicated that the dHex (Fuc) residue was linked to the Hex (Gal) and not to the HexNAc (GlcNAc) residue.

The online version of this article includes the following figure supplement(s) for figure 3:

**Figure supplement 1.** Preliminary characterization of TbFUT1 reaction products.

**Figure supplement 2.** Purification of the TbFUT1 reaction product by normal phase HPLC.

To obtain direct evidence, we performed a large-scale activity assay using LNB-O-Me as an acceptor and purified the reaction product by normal phase HPLC. Fractions containing the putative fucosylated trisaccharide product (*Figure 3—figure supplement 2*) were pooled and analysed for their neutral monosaccharide content, which showed the presence of Fuc, Gal, and GlcNAc. The purified reaction products were permethylated and analysed by electrospray ionisation mass spectrometry (ESI-MS) (*Figure 3A*), which confirmed that the main product was a trisaccharide of composition $dHex_1Hex_1HexNAc_1$. The MS/MS spectrum was also consistent with the dHex residue being attached to the Hex, rather than HexNAc, residue (*Figure 3B*). Subsequently, partially methylated alditol acetates (PMAAs) were generated from the purified trisaccharide product and analysed by gas chromatography coupled to mass spectrometry (GC-MS). This analysis identified derivatives consistent with the presence of non-reducing terminal-Fuc, 2-*O*-substituted Gal, and 3-*O*-substituted GlcNAc (*Figure 4—figure supplement 1* and *Table 2*), consistent with Fuc being linked to position 2 of Gal. The GC-MS methylation linkage analysis also revealed a trace of 2-*O*-substituted Fuc in the sample, which, together with the observation that 3′-FL, can act as a weak substrate (*Figure 2*, lane 18, and *Table 1*), may suggest that TbFUT1 can also form Fucα1,2Fuc linkages.

The purified TbFUT1 reaction product was also exchanged into deuterated water ($^2H_2O$) and analysed by one-dimensional $^1$H-NMR and two-dimensional $^1$H-ROESY (Rotating frame Overhouser Effect SpectroscopY). The $^1$H-NMR spectrum showed a doublet at about 5.1 ppm, consistent with the signal from the proton on the anomeric carbon ($H_1$) of an α-Fuc residue (*Figure 4A*).

A characteristic doublet for the anomeric proton of a β-Gal residue was also observed at 4.5 ppm. In the $^1$H-ROESY spectrum, a cross-peak (labelled

**Table 2.** Partially methylated alditol acetates (PMAAs) derivatives identified by GC-MS methylation linkage analysis of the purified TbFUT1 reaction product.

| PMAA derivative | RT (min) | Origin |
|---|---|---|
| 4,6-di-O-methyl-1,3,5-tri-O-acetyl-(1–2 H)- 2-N-methylacetamidoglucosaminitol | 24.6 | 3-*O*-substituted GlcNAc |
| 2,3,4,6-tetra-O-methyl-1,5-di-O-acetyl-(1–2 H)-galactitol | 16.7 | Non-reducing terminal Gal |
| 3,4,6-tri-O-methyl-1,2,5-tri-O-acetyl-(1–2 H)-galactitol | 18.6 | 2-*O*-substituted Gal |
| 2,3,4-tri-O-methyl-1,5-di-O-acetyl-(1–2 H)-fucitol | 14.1 | Non-reducing terminal Fuc |
| 3,4-di-O-methyl-1,2,5-tri-O-acetyl-(1–2 H)-fucitol | 15.9 | 2-*O*-substituted Fuc |

RT: retention time.

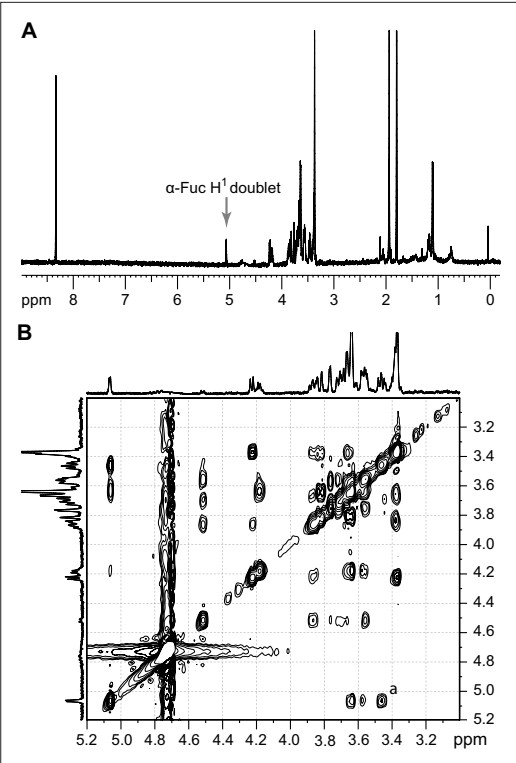

**Figure 4.** $^1$H-NMR and $^1$H-ROESY spectra of the TbFUT1 reaction product. (**A**) One-dimensional $^1$H-NMR spectrum. The *arrow* points to the α-Fuc $H_1$ doublet. (**B**) Enlargement of the 3.2–5.1 ppm region of the two-dimensional $^1$H-ROESY spectrum. (**a**) indicates the crosspeak resulting from a through-space connectivity between α-Fuc $H_1$ and β-Gal $H_2$.

The online version of this article includes the following figure supplement(s) for figure 4:

**Figure supplement 1.** GC-MS methylation linkage analysis of the purified reaction product.

a) could be observed indicating a through-space connectivity between the $H_1$ of α-Fuc and the $H_2$ of a β-Gal, consistent with a Fucα1,2Gal linkage in the TbFUT1 reaction product (***Figure 4B***). The chemical shifts that could be clearly assigned by either one-dimensional $^1$H-NMR or two-dimensional $^1$H-ROESY are listed in ***Table 3***.

Taken together, these data unambiguously define the structure of the TbFUT1 reaction product with GDP-Fuc and LNB-O-Me as Fucα1,2Galβ1,3GlcNAcβ1-O-$CH_3$ which, in turn, defines TbFUT1 as having a GDP-Fuc: β-Gal α-1,2-fucosyltransferase activity with an apparent preference for a Galβ1,3GlcNAcβ1-O-R acceptor motif.

## Generation of TbFUT1 conditional null mutants in procyclic and bloodstream form *T. brucei*

Semi-quantitative RT-PCR showed that *TbFUT1* mRNA was present in both bloodstream form and procyclic form *T. brucei* (data not shown). We therefore sought to explore TbFUT1 function in both lifecycle stages by creating *TbFUT1* conditional null mutants. The strategies used to generate the mutants are described in ***Figure 5***. The creation of these mutants was possible because genome assembly indicated *TbFUT1* to be present as a single copy per haploid genome, and Southern blot analysis using a *TbFUT1* probe was consistent with this prediction (***Figure 5—figure supplement 1***). In procyclic cells (***Figure 5***, left panel), the first *TbFUT1* allele was replaced by homologous recombination with linear DNA containing the puromycin resistance gene (*PAC*) flanked by about 500 bp of the *TbFUT1* 5′- and 3′-UTRs. After selection with puromycin, an ectopic copy of *TbFUT1*, under the control of a tetracycline-inducible promoter, was introduced in the ribosomal DNA (rDNA) locus using phleomycin selection. Following induction with tetracycline, the second allele was replaced with the *BSD* gene by homologous recombination, generating the final procyclic form *ΔTbFUT1::PAC/TbFUT1^Ti^/ΔTbFUT1::BSD* conditional null mutant cell line (PCF *TbFUT1* cKO). In bloodstream form cells (***Figure 5***, middle panel), an ectopic copy of *TbFUT1* carrying a C-terminal MYC$_3$ epitope tag under the control of a tetracycline-inducible promoter was first introduced into the ribosomal DNA (rDNA) locus using phleomycin selection. Following cloning and induction with tetracycline, the first *TbFUT1* allele was

**Table 3.** $^1$H-NMR and $^1$H-ROESY chemical shift assignments for the purified TbFUT1 reaction product.

| Residue | H1 | H2 | H3 | H4 | H5 | H6/6' | NAc |
|---------|-----|-----|-----|-----|-----|--------|-----|
| αFuc | 5.05 (J = 4 Hz) | 3.57 | 3.67 | 3.63 | 4.2 | 1.1 | |
| βGal | 4.5 | 3.45 | 3.55 | 3.89 | ND | ND | |
| βGlcNAc | ND | 3.63 | ND | 3.4 | ND | 3.78/3.89 | 2.1 |

J: coupling constant. ND: chemical shift could not be clearly assigned.

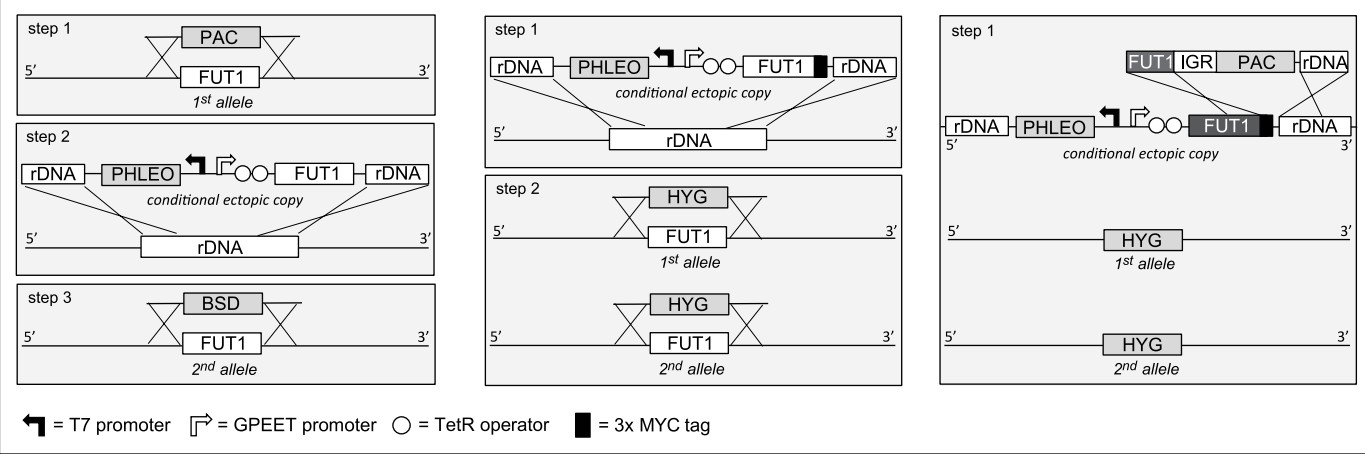

**Figure 5.** Cloning strategies for the creation of the *TbFUT1* conditional null mutants. *Left panel*: To create the procyclic form conditional null mutant (PCF *TbFUT1* cKO), the first *TbFUT1* allele was replaced by *PAC*, an ectopic tetracycline-inducible copy of the *TbFUT1* gene was introduced into the ribosomal DNA locus, and the second *TbFUT1* allele was replaced by *BSD*. *Middle panel*: To create the bloodstream form conditional null mutant (BSF *TbFUT1-MYC₃* cKO), an ectopic tetracycline-inducible copy of the *TbFUT1* gene with a MYC₃ tag was first introduced into the ribosomal DNA locus. Both *TbFUT1* alleles were subsequently replaced by *HYG* through homologous recombination followed by gene conversion. *Right panel:* To create the untagged bloodstream form cKO (BSF *TbFUT1* cKO), the BSF *TbFUT1-MYC₃* cKO mutant (*middle panel*) was modified by homologous recombination with a construct that removed the C-terminal MYC₃ tag under *PAC* selection. IGR: intergenic region.

The online version of this article includes the following figure supplement(s) for figure 5:

**Figure supplement 1.** Southern blots of *TbFUT1* conditional null mutants.

then targeted for homologous recombination with linear DNA containing the hygromycin resistance gene (*HYG*) flanked by about 1200 bp of the *TbFUT1* 5′- and 3′-UTRs. After selection with hygromycin, Southern blotting revealed that gene conversion had taken place and that both *TbFUT1* alleles had been replaced by *HYG* yielding a bloodstream form $TbFUT1\text{-}MYC_3{}^{Ti}/\Delta TbFUT1::HYG/\Delta TbFUT1::HYG$ conditional null mutant cell line (BSF *TbFUT1*-MYC₃ cKO). Southern blotting data confirming the genotypes of these mutants are shown in **Figure 5—figure supplement 1**. The BSF cell line was also used to generate a $TbFUT1^{Ti}/\Delta TbFUT1::HYG/\Delta TbFUT1::HYG$ conditional null mutant cell line by in situ homologous recombination of the tetracycline inducible *TbFUT1*-MYC₃ copy, converting it to an untagged *TbFUT1* gene and generating BSF *TbFUT1* cKO (**Figure 5**, right panel).

## TbFUT1 is essential to procyclic and bloodstream form *T. brucei*

Procyclic and bloodstream form *TbFUT1* conditional null mutants were grown under permissive (plus tetracycline) or non-permissive (minus tetracycline) conditions. The PCF *TbFUT1* cKO cells grown under non-permissive conditions showed a clear reduction in the rate of cell growth after 6 days, eventually dying after 15 days (**Figure 6A**).

The BSF *TbFUT1* cKO cells grew like wild-type cells under permissive conditions, whether or not the expressed TbFUT1 had a C-terminal MYC₃ tag, and under non-permissive conditions also showed a clear reduction in the rate of cell growth after 2–4 days, dying after 3–5 days (**Figure 6B,C**). These growth phenotypes are very similar to those described for procyclic and bloodstream form *TbGMD* conditional null mutants that cannot synthesize GDP-Fuc under non-permissive conditions (**Figure 6— figure supplement 1**; **Turnock et al., 2007**). This is consistent with the hypothesis that TbFUT1 may be the only enzyme that utilizes GDP-Fuc, or at least that it is the only FUT transferring fucose to essential acceptors. Further evidence that TbFUT1 is essential for procyclic and bloodstream form growth was obtained from northern blots (**Figure 6D,E**). These show that *TbFUT1* mRNA levels are undetectable for several days after the removal of tetracycline, but that growth resumes only when some cells escape tetracycline control after about 29 days (procyclic form) and 11 days (bloodstream form). Escape from tetracycline control after several days is typical of conditional null mutants for essential trypanosome genes (**Roper et al., 2002**). Evidence for the expression of the MYC₃ tagged TbFUT1 protein in the BSF *TbFUT1*-MYC₃ cKO cell line and of unmodified TbFUT1 in the BSF *TbFUT1* cKO cell line under permissive conditions is shown in **Figure 6F**.

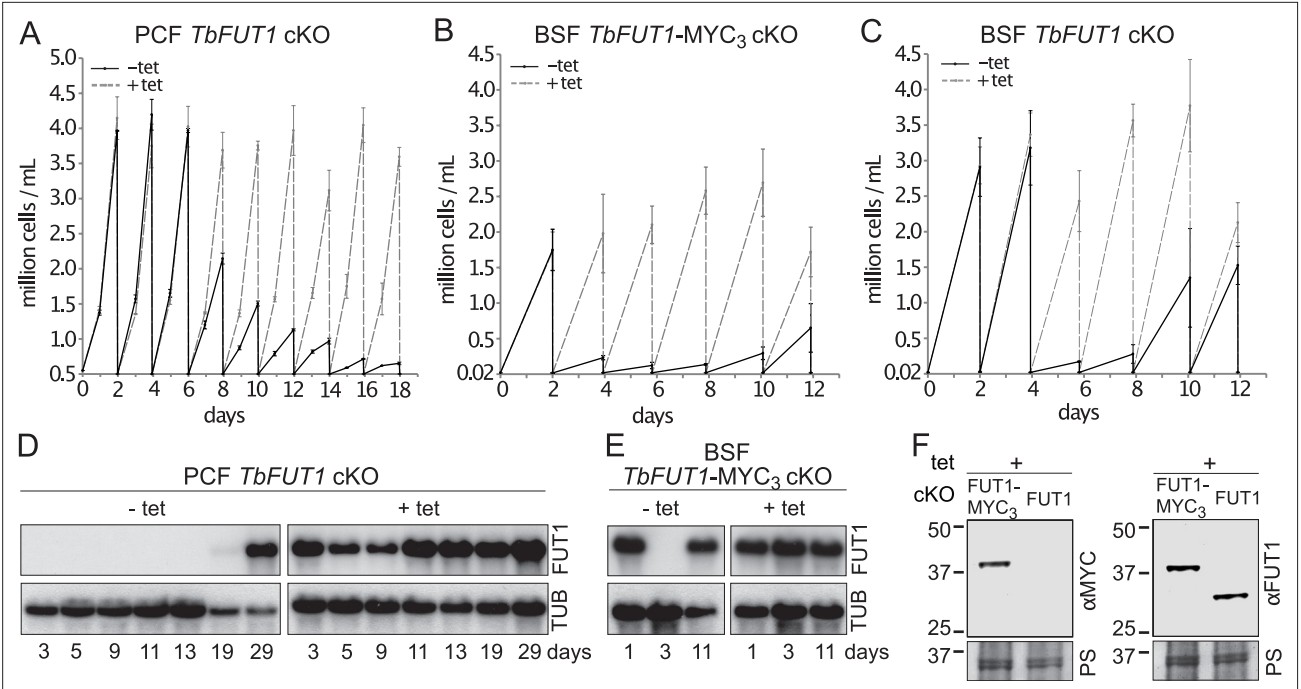

**Figure 6.** TbFUT1 is essential for procyclic and bloodstream form cell growth in vitro. The cell numbers (± standard deviation) for TbFUT1 cKO under permissive (plus tetracycline, *dotted line*) and non-permissive (minus tetracycline, *solid line*) conditions are shown for three procyclic (**A**) and bloodstream form (**C**) clones, as well as for three bloodstream clones carrying a tetracycline-inducible ectopic *TbFUT1* gene with a C-terminal MYC₃ tag (**B**). For each clone procyclic form clone, two biological repeats were analysed and three for bloodstream form clones. (**D, E**) Corresponding *TbFUT1* mRNA levels were determined by northern blots. Alpha-tubulin (TUB) was used as a loading control. (**F**) TbFUT1-MYC₃ and untagged TbFUT1 are detected by western blot analysis in the respective bloodstream form cKO cell lines under permissive conditions (+Tet). The *left panel* shows an anti-MYC (αMYC) blot and the *right panel* an anti-recombinant TbFUT1 antibody (αFUT1) blot. Membranes were stained with Ponceau S (PS) to ensure equal loading.

The online version of this article includes the following figure supplement(s) for figure 6:

**Figure supplement 1.** Comparison of *TbFUT1* and *TbGMD* conditional null mutant growth.

**Figure supplement 2.** *TbFUT1* and *TbGMD* cKO have increased cell size.

**Figure supplement 3.** The paraflagellar rod (PFR) and flagellar attachment zone (FAZ) appear normal in PCF *TbFUT1* cKO.

**Figure supplement 4.** *TbFUT1* and *TbGMD* cKO procyclic form cells show no defect in motility.

From a morphological point of view, both procyclic form *TbGMD* and *TbFUT1* conditional null mutants grown under non-permissive conditions showed an increase in average cell volume, due to increased cell length, concomitant with the start of the cell growth phenotype (*Figure 6—figure supplement 2A*). However, we were unable to reproduce the flagellar detachment phenotype previously reported for the PCF *TbGMD* cKO grown in non-permissive conditions (*Turnock et al., 2007*), nor was such a phenotype observed in the PCF *TbFUT1* cKO parasites, either by scanning electron microscopy or immunofluorescence (IFA) (*Figure 6—figure supplement 2C* and *Figure 6—figure supplement 3*). The percentage of cells displaying flagellar detachment (1.5–2%) in both null mutants grown in non-permissive conditions (*Figure 6—figure supplement 2B*) was consistent with what has previously been reported for wild-type cells (*LaCount et al., 2002*). Additionally, we could observe no defect in cell motility in either PCF *TbGMD* or PCF *TbFUT1* cKO grown in non-permissive conditions (*Figure 6—figure supplement 4*).

## TbFUT1 localizes to the parasite mitochondrion

The BSF *TbFUT1*-MYC₃ cKO cell line, grown under permissive conditions, was stained with anti-MYC antibodies and produced a pattern suggestive of mitochondrial localization. This was confirmed by co-localization with MitoTracker (*Figure 7A*). However, when TbFUT1 was introduced into wild-type cells fused with an HA₃ epitope tag at the N-terminus, either with or without a C-terminal MYC₃-tag

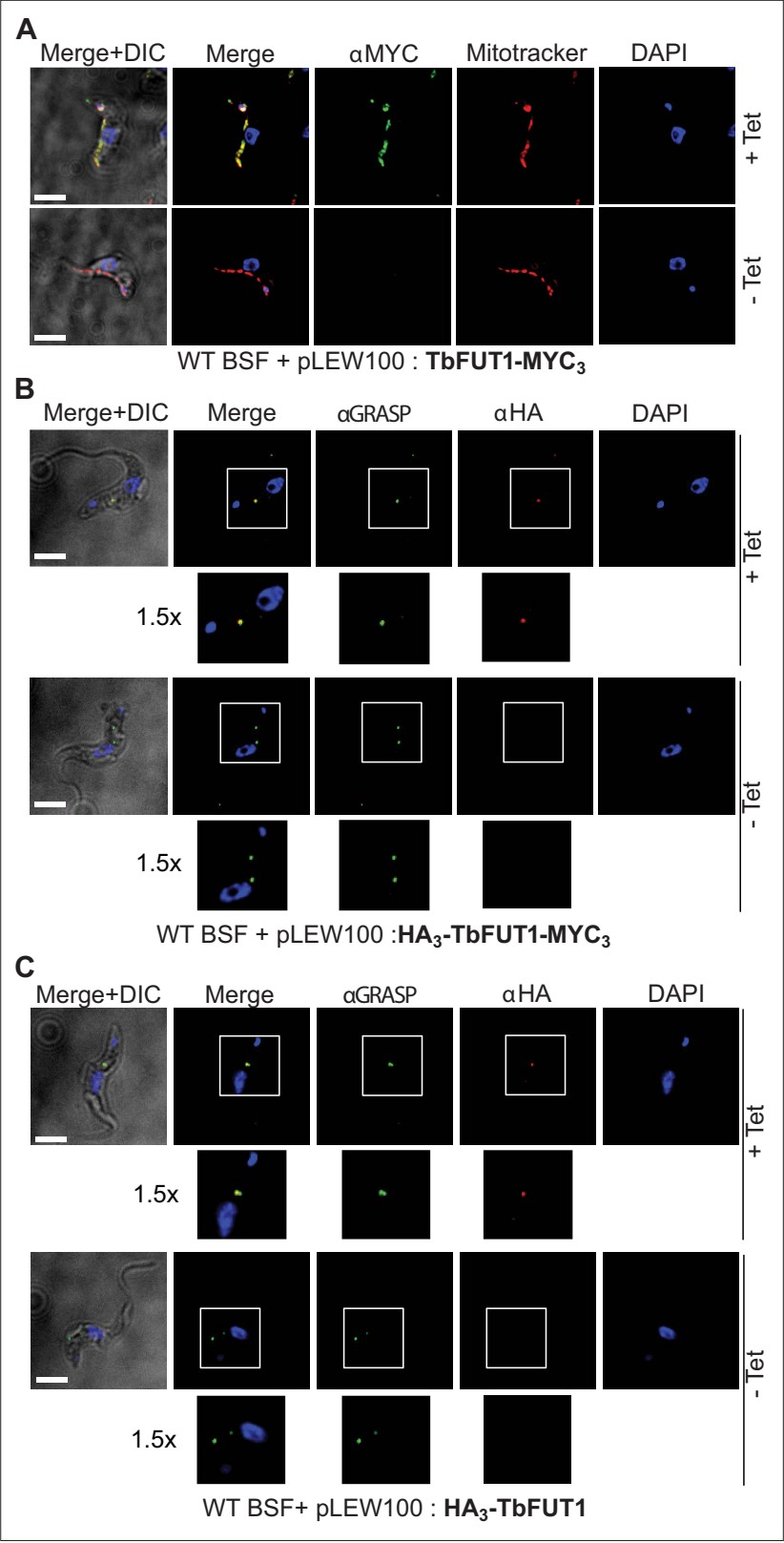

**Figure 7.** C- and N-terminal tagging of TbFUT1 result in mitochondrial and Golgi apparatus localization, respectively. (**A**) Bloodstream form (BSF) cKO parasites expressing tet-inducible C-terminally tagged TbFUT1-MYC$_3$ were imaged under permissive (+Tet) and non-permissive (-Tet) conditions by differential interference contrast (DIC) and fluorescence microscopy after staining with anti-MYC, MitoTracker, and DAPI. Comparable patterns were

*Figure 7 continued on next page*

*Figure 7 continued*

observed for anti-MYC and MitoTracker, suggesting that TbFUT1-MYC$_3$ localizes to the mitochondrion. (**B, C**) IFA of BSF cKO parasites expressing a tet-inducible N-terminally tagged HA$_3$-TbFUT1-MYC$_3$ (**B**) or HA$_3$-TbFUT1 (**C**) after labelling with anti-HA, anti-GRASP, and DAPI suggests a Golgi apparatus location for both HA$_3$-TbFUT1-MYC$_3$ and HA$_3$-TbFUT1. The absence of anti-MYC (**A**) or anti-HA (**B, C**) staining under non-permissive conditions confirms the specificity of the labelling for the respective TbFUT1 fusion proteins. *White boxes*: areas magnified 1.5× in the rows below. Scale bars: 3 µm.

(constructs pLEW100:HA$_3$-FUT1-MYC$_3$ and pLEW100:HA$_3$-FUT1), the tagged protein co-localized with GRASP, a marker of the Golgi apparatus (***Figure 7B,C***). In these cases, we suspect that N-terminal tagging has disrupted mitochondrial targeting by obscuring the N-terminal mitochondrial targeting sequence. Indeed, no mitochondrial targeting motif was predicted in silico for N-terminal HA$_3$ tagged TbFUT1. Nevertheless, since the mitochondrial localization of a FUT is unprecedented, we elected to raise polyclonal antibodies against recombinant TbFUT1 to further assess its subcellular location. To do so, an N-terminally His$_6$-tagged Δ$_{32}$TbFUT1 protein was expressed, re-solubilized from inclusion bodies and used for rabbit immunization. The IgG fraction was isolated on immobilized protein-A

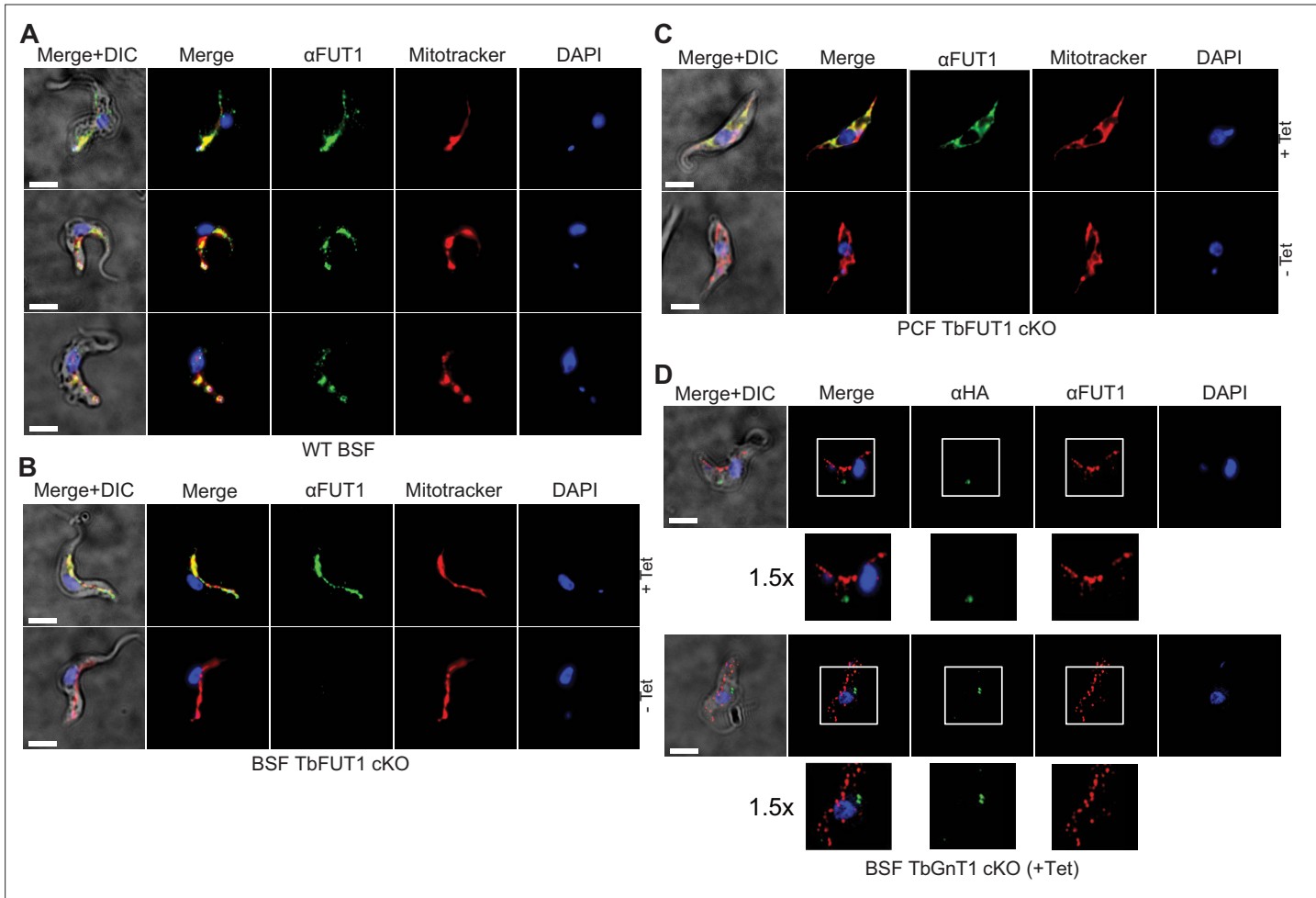

**Figure 8.** Antibodies to the recombinant protein localize TbFUT1 to the mitochondrion. (**A**) IFA of wild-type bloodstream form (BSF) trypanosomes after staining with affinity-purified anti-TbFUT1 (αFUT1) MitoTracker and DAPI. Comparable patterns were observed for αFUT1 and MitoTracker, suggesting that TbFUT1 localizes to the mitochondrion. (**B, C**) *TbFUT1* conditional null mutants were grown under permissive (+Tet) and non-permissive (-Tet) conditions for 3 days in the case of BSF (**B**) or 7 days for PCF (**C**) and imaged.. In both cases, the tetracycline-inducible TbFUT1 pattern is consistent with a mitochondrial localization. (**D**) BSF trypanosomes induced to express a C-terminally tagged known Golgi glycosyltransferase (TbGnTI-HA$_3$) were imaged after staining with αFUT1, anti-HA, and DAPI, as indicated. The merged images of two representative cells suggest no significant co-localization between native TbFUT1 and the Golgi-localized TbGnT1. *White boxes*: areas magnified 1.5× in the rows below. Scale bars: 3 µm.

and the anti-TbFUT1 IgG sub-fraction affinity purified on immobilized recombinant GST-TbFUT1 fusion protein. To further ensure mono-specificity of the antibodies to TbFUT1, the resulting fraction was adsorbed against a concentrated cell lysate of the PCF *TbFUT1* cKO mutant grown for 9 days under non-permissive conditions. The resulting highly specific polyclonal antibody was used to detect TbFUT1 expression in wild-type bloodstream form cells as well as in BSF and PCF *TbFUT1* cKO cells under permissive and non-permissive conditions (*Figure 8A–C*).

Anti-TbFUT1 antibodies co-localized with MitoTracker staining in the wild-type cells and in the conditional null mutants under permissive conditions. No signal for the anti-TbFUT1 antibodies was seen under non-permissive conditions, confirming the specificity of the polyclonal antibody. Taking the possibility of a dual Golgi/mitochondrial localization into account, TbFUT1 localization was also assessed in bloodstream form cells ectopically expressing TbGnTI-HA₃ as an authentic Golgi marker (*Damerow et al., 2014*). No co-localization between TbGnTI-HA₃ and anti-TbFUT1 was observed, suggesting that TbFUT1 is either exclusively or predominantly expressed in the parasite mitochondrion (*Figure 8D*).

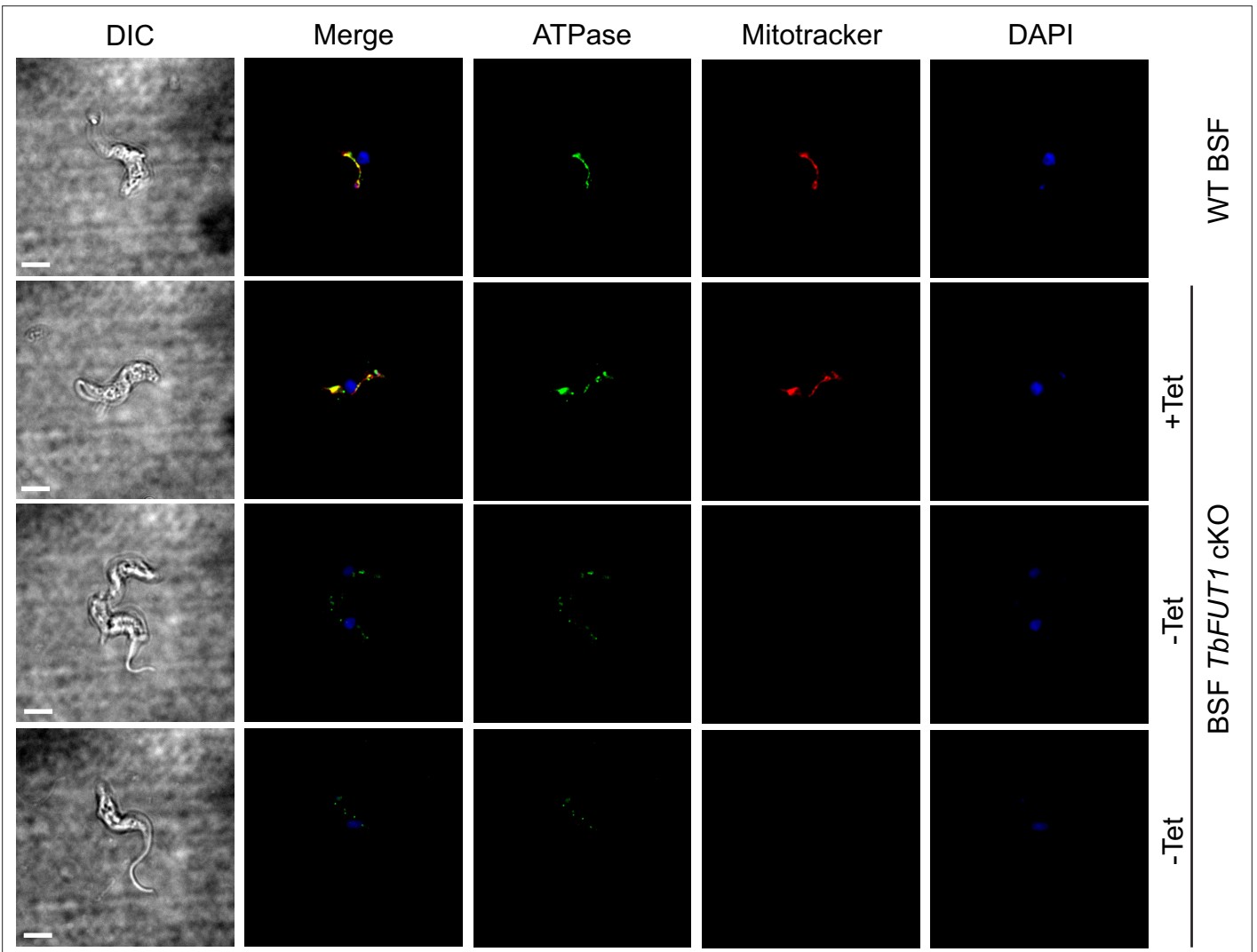

**Figure 9.** Absence of TbFUT1 disturbs mitochondrial activity. Bloodstream form (BSF) wild-type and *TbFUT1* cKO parasites were cultured for 5 days under permissive (+Tet) and non-permissive (-Tet) conditions, fixed and labelled with MitoTracker for mitochondrial potential and with anti-mitochondrial ATPase antibody. In mutants grown in non-permissive conditions (*lower panels*), both ATPase and MitoTracker staining are strongly reduced, suggesting reduced mitochondrial functionality. Scale bar: 3 μm.

The online version of this article includes the following figure supplement(s) for figure 9:

**Figure supplement 1.** Absence of TbFUT1 disturbs mitochondrial activity.

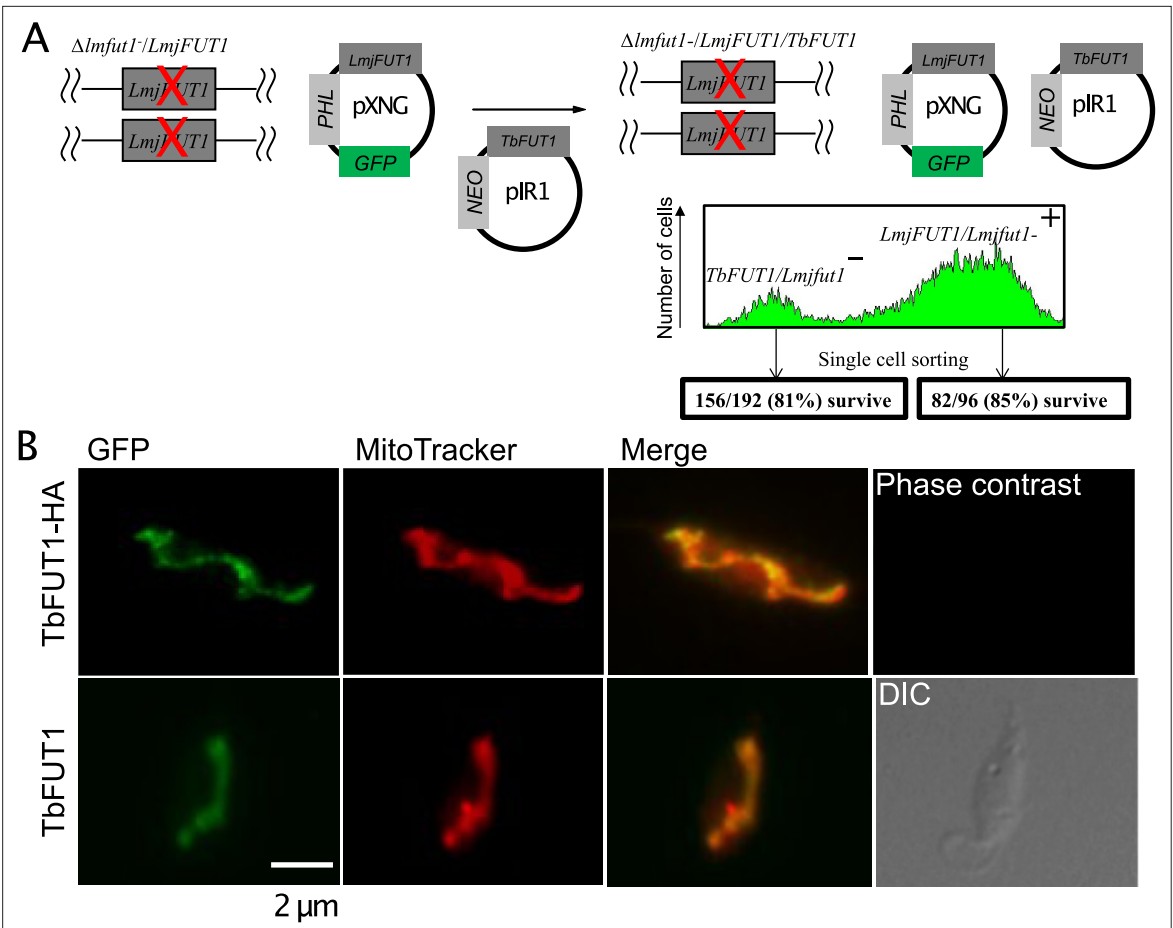

**Figure 10.** TbFUT1 can functionally and spatially replace LmjFUT1. (**A**) Outline of the 'plasmid shuffling' procedure to replace *LmjFUT1* with *TbFUT1*. An *L. major* homozygous *fut1-* null mutant expressing *LmjFUT1* (from the GFP-positive, episomal pXNGPHLEO vector) was transfected with pIR1NEO-*TbFUT1* (±HA) expressing either WT or HA-tagged TbFUT1 (GFP-negative). Following transfection and growth in the absence of selective drugs, single cells were sorted for GFP expression or loss. A similar yield of colonies was obtained from both populations. (**B**) TbFUT1 HA-tagged (top row) or WT (bottom row) was expressed in an *L. major* chromosomal *fut1-* mutant background and analysed by indirect immunofluorescence using anti-HA or anti-TbFUT1 antibodies. MitoTracker red was used as a mitochondrial marker. Scale bar: 2 μm.

## Deletion of TbFUT1 disturbs mitochondrial activity

Bloodstream form wild-type and BSF *TbFUT1* cKO cells, grown with and without tetracycline for 5 days, were stained with antibodies to mitochondrial ATPase and with MitoTracker. As expected, ATPase and MitoTracker co-localized in wild-type cells and in the mutant under permissive conditions (*Figure 9*, top panels). However, under non-permissive conditions the few remaining viable cells showed significantly diminished MitoTracker staining, indicative of reduced mitochondrial membrane potential, and a reduction in ATPase staining, suggesting that TbFUT1 is in some way required for mitochondrial function (*Figure 9*, lower panels, and *Figure 9—figure supplement 1*).

## TbFUT1 can replace the homologous essential mitochondrial fucosyltransferase in *L. major*

The essentiality and mitochondrial location of the orthologous protein in *L. major*, LmjFUT1, was reported in *Guo et al., 2021*. Here we asked whether TbFUT1 could functionally substitute for LmjFUT1 using a plasmid shuffle approach (*Guo et al., 2017*; *Murta et al., 2009*). We started with an *L. major* homozygous *fut1-* null mutant expressing *LmjFUT1* from the episomal pXNGPHLEO vector which additionally expresses GFP (*Guo et al., 2017*; *Murta et al., 2009*) and transfected it with an episomal construct pIR1NEO-*TbFUT1* (±HA) expressing WT or HA-tagged *TbFUT1*. Following transfection, parasites were grown briefly in the absence of any selective drug, and single cells were sorted

for GFP expression and selected for phleomycin resistance, or GFP loss accompanied by G418 (*NEO*) resistance (*Figure 10A*). In these studies, 85% (82/96) of the GFP-positive (GFP+) clonal lines (positive control) grew when placed in microtiter wells, while 81% (156/192) of the GFP-negative (GFP-) sorted cells likewise grew (*Figure 10A*). Control PCR studies confirmed the loss of *TbFUT1* in the GFP- clones (not shown). The similar survival of cells bearing *Lmj* or *TbFUT1* provides strong evidence that the TbFUT1 can fully satisfy the *L. major* FUT1 requirement (*Murta et al., 2009*).

Immunofluorescence microscopy of progeny *Lmj fut1-/*pIR1NEO-TbFUT1 (±HA)-transfected parasites with either mouse monoclonal anti-HA antibodies or affinity-purified rabbit polyclonal anti-TbFUT1 antibodies showed that TbFUT1 was faithfully localized in the *Leishmania* mitochondrion (*Figure 10B*). The Pearson *r* correlation for co-localization of the anti-HA or anti-TbFUT1 signals with MitoTracker red was >0.98 in both experiments. In a negative control where anti-TbFUT1 antibody was omitted, the fluorescence signal was lost (data not shown). This result establishes conservation of mitochondrial targeting of TbFUT1 when heterologously expressed in *L. major* promastigotes.

## Discussion

The presence and essentiality of the nucleotide sugar donor GDP-Fuc in BSF and PCF trypanosomes led us to search for putative FUTs in the parasite genome. Only one gene (*TbFUT1*; Tb927.9.3600), belonging to the CAZy GT11 family, was found, and phylogenetic analyses revealed that one, or two in the case of *T. cruzi*, orthologue could be found in the genomes of other kinetoplastids. These putative kinetoplastid FUTs form a distinct clade within the GT11 FUT superfamily and are distinct from the GT10 FUTs found in *T. cruzi, L. major,* and related parasites, which are absent in *T. brucei*. Consistent with TbFUT1 being the only enzyme likely to utilize the essential metabolite GDP-Fuc, we found that TbFUT1 is also essential to both BSF and PCF parasites. We were able to express TbFUT1 in *E. coli,* and the recombinant enzyme was used to demonstrate its activity as a GDP-Fuc: β-Gal α-1,2-fucosyltransferase with an apparent preference for a Galβ1,3GlcNAcβ1-O-R acceptor motif out of the acceptor substrates investigated.

Immunofluorescence analysis, using an affinity-purified antibody raised against native TbFUT1 as well as C-terminal epitope tagging, suggests that TbFUT1 is located in the parasite mitochondrion. Consistent with this, TbFUT1 was identified in the mitochondrial proteome of PCF *T. brucei* (*Panigrahi et al., 2009*). Although this was based on a single unique peptide, and is therefore not a high-confidence assignment, this is the only proteomic dataset deposited on TriTrypDB (*Aslett et al., 2010*) that identified TbFUT1. A quantitative proteomic analysis that estimated protein turnover in *T. brucei* (*Tinti et al., 2019*), and ranking of protein abundances using those deep proteomic data sets (*Ji et al., 2021*), did not identify TbFUT1 in either BSF or PCF, suggesting that it is a very low-abundance protein. The immunofluorescence analysis performed in this study does not allow us to define the specific localization of TbFUT1 in the parasite mitochondrion. Generally, proteins that are targeted to the mitochondrion via an *N*-terminal presequence are translocated across both the outer- and inner-mitochondrial membranes into the mitochondrial matrix where the presequence is proteolytically cleaved (*Long et al., 2008*). Since TbFUT1 has a predicted mitochondrial targeting sequence (*Figure 1A*), we speculate that TbFUT1 may be located in the mitochondrial matrix. However, we are mindful that the putative presequence also overlaps with the predicted FUT motif IV (*Li et al., 2008*). Future work will address whether the FUT is indeed imported to the mitochondrial matrix, the intermembrane space, or the organelle outer membrane and will include the functional characterization of the role of the predicted TbFUT1 mitochondrial targeting sequence in protein localization. These studies will also assess the fate of motif IV and its function in TbFUT1 enzymology.

Regardless of the precise sub-organellar localization, the localization of TbFUT1 to the mitochondrion is a highly unusual result. Although in recent years fucosylation has been described in the nucleus and cytosol of protists and plants (*Bandini et al., 2016*; *Rahman et al., 2016*; *Van Der Wel et al., 2002*; *Zentella et al., 2017*), we are unaware of any other examples of mitochondrial FUTs in any organism. Mitochondrial glycosylation in general is poorly understood. The only other known example of a mitochondrial-localized glycosyltransferase is mitochondrial *O*-GlcNAc transferase (mOGT) (*Bond and Hanover, 2015*). Mitochondria of rat cardiomyocytes are also positive for OGA and express a UDP-GlcNAc transporter on their outer membrane, indicating all of the molecular components required for this cycling post-translational modification are present in the organelle (*Banerjee et al., 2015*). Interestingly, disruption of mOGT in HeLa cell mitochondria also leads to

mitochondrial dysfunction. While there are no (m)OGT orthologues in kinetoplastids, these observations highlight some of the challenges inherent for a mitochondrial-localized glycosyltransferase. Firstly, for a TbFUT1 that localizes to either the intermembrane space or the mitochondrial matrix to be active, GDP-Fuc would need to be imported into the mitochondrion, which would require the presence of an uncharacterized mitochondrial GDP-sugar transporter. Secondly, TbFUT1 appears to be an α-1,2-FUT that decorates glycans terminating in Galβ1,3GlcNAc, suggesting either that additional uncharacterized glycosyltransferases and nucleotide sugar transporters may be present in the parasite mitochondrion, or that a glycoconjugate substrate may be assembled in the cytoplasm or secretory pathway and then somehow translocated to or imported into the mitochondrion to be modified by TbFUT1. Experiments to resolve these options will be undertaken, as will further experiments to try to find the protein, lipid, and/or other acceptor substrates of TbFUT1. The latter may then provide clues as to why TbFUT1 is essential for mitochondrial function and parasite growth. Several attempts to identify TbFUT1 substrates have failed so far, including fucose-specific lectin blotting and pull-downs, [$^3$H]fucose labelling of parasites transfected with GDP-Fuc salvage pathway enzymes, and LC-MS/MS precursor ion and neutral-loss scanning methods. Although the significance is unclear, it is interesting to note that procyclic TbFUT1 has been shown to be under circadian regulation (*Rijo-Ferreira et al., 2017*).

In conclusion, TbFUT1 is an essential -1,2-FUT that localizes to either outer membrane or inner compartments of the parasite mitochondrion and presents orthologues throughout the kinetoplastida. As described in *Guo et al., 2021*, both activity and mitochondrial localization of the *L. major* homologue are required for parasite viability, whereas other putative *L. major* FUTs targeted to the secretory pathway are dispensable. Although no data are available so far on the enzymes from *T. cruzi* or other members of this group, these initial results suggest the intriguing possibility of an essential, conserved mitochondrial fucosylation pathway in kinetoplastids that might be exploitable as a common drug target.

# Materials and methods

**Key resources table**

| Reagent type (species) or resource | Designation | Source or reference | Identifiers | Additional information |
|---|---|---|---|---|
| Gene (*Trypanosoma brucei*) | *TbFUT1* | Genedb.org | Tb927.9.3600 | |
| Strain, strain background (*Trypanosoma brucei*) | Strain 427, clone 29.13 (PCF WT) | *Wirtz et al., 1999* | | Procyclic form |
| Strain, strain background (*Trypanosoma brucei*) | Strain 427, variant MITaT 1.2 (BSF WT) | *Wirtz et al., 1999* | | Bloodstream form |
| Strain, strain background (*Leishmania major*) | *L. major* Friedlin V1 | *Guo et al., 2017* | MHOM/IL/80/Friedlin | |
| Strain, strain background (*Escherichia coli*) | BL21 (DE3) | Agilent | | |
| Strain, strain background (*Escherichia coli*) | BL21 Gold(DE3) | Agilent | | |
| Recombinant DNA reagent | pLEW100 | Prof. George Cross (Rockefeller University) | Addgene Plasmid #24011 | (Plasmid) |
| Recombinant DNA reagent | pMOTag4YH-PAC | Prof. Thomas Seebeck (University of Bern); *Oberholzer et al., 2006* | | (Plasmid) |
| Recombinant DNA reagent | pIR1NEO | *Guo et al., 2021* | | (Plasmid) |
| Recombinant DNA reagent | pGEX6P1-GST-PP | Prof. Daan Van Aalten (University of Dundee) | | (Plasmid) |
| Recombinant DNA reagent | pET15b | Novagen | | (Plasmid) |

*Continued on next page*

*Continued*

| | | | |
|---|---|---|---|
| Chemical compound, drug | GDP[³H]Fuc | American Radiochemicals | |
| Chemical compound, drug | Galβ1,3GlcNAcβ-OMe (LNB-O-Me) | Toronto Research Chemicals | |
| Peptide, recombinant protein | *X. manihotis* α–1,2-fucosidase | New England Biolabs | |
| Commercial assay or kit | GDP-Glo Kit | Promega | |
| Other | MitoTracker Red CMX Ros | Thermo Fisher | |
| Antibody | (Rabbit polyclonal) anti-TbFUT1 IgG | This study: antibody generated as described in Materialsandmethods | (1:1000) |
| Antibody | (Rabbit polyclonal) anti-GRASP IgG | Prof. Graham Warren (University College London)/Prof. Chris de Graffenreid (Brown University) | (1:1000) |
| Antibody | (Rabbit polyclonal) anti-ATPase IgG | Prof. David Horn (University of Dundee) | (1:1000) |
| Antibody | (Mouse polyclonal) anti-FAZ IgM | Prof. Keith Gull (University of Oxford) | (1:2) |
| Antibody | (Mouse polyclonal) anti-PFR IgG | Prof. Keith Gull (University of Oxford) | (1:10) |

## Parasite strains

*T. brucei* procyclic form (strain 427, clone 29.13) and bloodstream form (strain 427, variant MITaT 1.2) were used in these experiments. Both strains are stably expressing T7 polymerase and tetracycline repressor protein under G418 (bloodstream) or G418 and hygromycin (procyclic) selection (*Wirtz et al., 1999*). Procyclic form *T. brucei* was cultured in SDM-79 medium (*Brun and Schönenberger, 1979*) containing 15% fetal bovine serum and GlutaMAX at 28 °C. Bloodstream form *T. brucei* was grown in HMI-9t at 37 °C, 5% $CO_2$. Induction was performed at 0.5 µg/ml tetracycline in both forms.

## BLASTp searches

The *T. brucei, T. cruzi,* and *L. major* predicted proteins (from the GeneDB database, http://www.genedb.org) were searched for putative FUTs using the BLASTp search algorithm (*Altschul et al., 1997*). The query input sequences are listed in *Supplementary file 1*. Protein sequence multiple alignments were assembled using Clustal (*Sievers et al., 2011*) and Jalview (*Waterhouse et al., 2009*).

## Cloning, protein expression, and purification of TbFUT1

The open reading frame (ORF) was amplified by PCR from *T. brucei* strain 427 genomic DNA and cloned into the N-terminal GST fusion vector pGEX-6P-1, modified to contain the PreScission Protease site (kind gift of Prof. Daan Van Aalten) using primers P1 and P2 (*Supplementary file 2*). Either the resulting pGEX6P1-GST-PP-TbFUT1 or pGEX6P1-GST-PP were transformed into BL21 (DE3) *E. coli* strain. Recombinant protein expression was induced with 0.1 mM isopropyl-β-D-thiogalactopyranoside (IPTG) and carried out at 16 °C for 16 hr. Prior to lysis by French press, cells were incubated for 30 min on ice in 50 mM Tris-HCl, 0.15 M NaCl, 1 mM DTT, pH 7.3 (Buffer A) with EDTA-free Complete Protease Inhibitors Tablet (Roche) and 1 mg/ml lysozyme. The soluble fraction, obtained by centrifugation at 17,000 × g, 4 °C for 30 min, was incubated 2 hr at 4 °C with Glutathione Sepharose Fast Flow beads that had been pre-equilibrated in Buffer A. The mixture was loaded into a disposable column and the beads washed first with 50 mM Tris-HCl, 0.25 M NaCl, 1 mM DTT, pH 7.3 (Buffer B), then with 50 mM Tris-HCl, 0.15 M NaCl, 0.1% sodium deoxycholate, 1 mM DTT, pH 7.3 (Buffer C), before being re-equilibrated in Buffer B. GST-TbFUT1 or GST were eluted in 50 mM Tris-HCl, 0.15 M NaCl, 10 mM reduced glutathione pH 8.0 (Buffer E). The yield of recombinant protein was estimated by Bradford assay and the degree of purification by SDS-PAGE analysis. Recombinant protein identification by peptide mass fingerprinting was performed by the Proteomic and Mass Spectrometry facility, School of Life Sciences, University of Dundee.

## Fucosyltransferase activity assays

### Radioactive assay

Aliquots of 2 µg of affinity-purified GST-TbFUT1 were incubated with 1 µCi GDP[³H]Fuc (American Radiochemicals), 1 mM acceptor in 50 mM Tris-HCl, 25 mM KCl, 5 mM MgCl₂, 5 mM MnCl₂, pH 7.2 for 2 hr at 37 °C. The acceptors tested (*Table 1*) were purchased from Sigma, Dextra Laboratories, or Toronto Research Chemicals. To study the dependency on divalent cations, MgCl₂ and MnCl₂ were removed from the buffer and a formulation with 10 mM EDTA was also tested. Reactions were stopped by cooling on ice then desalted on mixed bed as follows: 100 µl each Chelex100 (Na⁺) over Dowex AG50 (H⁺) over Dowex AG3 (OH⁻) over QAE-Sepharose A25 (OH⁻). The flow through and the elutions (4 × 400 µl of water) were combined. About 5% of the desalted reactions were counted at a LS 6500 scintillation counter (Beckmann). The remaining material was lyophilized for further analyses.

### Luminescence assay

Recombinant GST-TbFUT1 fusion protein was expressed in *E. coli* as described above, except that cell lysis was performed without lysozyme using a Cell Disruptor (Constant Systems) in place of a French press. After lysis, the suspension was centrifuged at 45,000× g, 30 min, 4 °C, and the fusion protein was purified using glutathione Sepharose 4B. SDS-PAGE analysis of the glutathione-eluted purified GST-TbFUT1 was like that shown in (*Figure 2—figure supplement 1*). The purified GST-TbFUT1 was diluted 1:10 into assay buffer and 20 µl aliquots containing about 0.1 µg of GST-TbFUT1 (and no-enzyme controls) were incubated with GDP-Fuc, GDP-Man, or GDP-Glc (final concentrations 50 mM Tris-HCl, pH 7.2, 25 mM KCl, 50 µM GDP-sugar), with and without 0.8 mM LNB, overnight at room temperature in 96-well flat-bottom luminometer plates (Corning Costar 3688). Reactions were stopped and developed with 25 µl Promega GDP-Glo reagent and after 1 hr luminescence was read on a BMG Labtech Pherostar plate reader. A calibration curve using 25 µl aliquots of GDP from 0 to 25 µM in assay buffer was developed in parallel. All measurements were performed in triplicate.

## HPTLC analysis

Reaction products and standards were dissolved in 20% 1-propanol and separated on a 10 cm HPTLC Si-60 plates (Merck) using 1-propanol:acetone:water 9:6:4 (v:v:v) as mobile phase. Non-radiolabelled sugars were visualized by orcinol/H₂SO₄ staining. In the case of radiolabelled products, the HPTLC plates were sprayed with En³hance (PerkinElmer) and visualized by fluorography.

### Recovery of radiolabelled products

The reaction products from the TbFUT1 activity assays using Lac, LNB, and LacNAc as acceptors were separated on a 20 cm HPTLC Si-60 plate. The radiolabelled products were localized on the HPTLC plate using an AR-2000 (BioScan) plate reader, the corresponding areas were wetted with mobile phase, and the silica scraped and transferred to microcentrifuge tubes. The radiolabelled materials were extracted from the solid phase by incubating with mobile phase and a 5% aliquot counted at the scintillation counter to determine the amount of recovered material.

## Acid hydrolysis

Aliquots corresponding to 15% of the extracted radiolabelled reaction products were dried in a Speedvac concentrator, re-dissolved in 4 M TFA, and incubated at 100 °C for 4 hr (*Ferguson, 1992*). After cooling, the samples were dried on a Speedvac concentrator, washed in water, and re-dissolved in 20% 1-propanol to be analysed by HPTLC as described in the main text.

### *Xanthomonas manihotis* α-1,2-fucosidase reaction

Aliquots corresponding to 15% of the extracted radiolabelled reaction products were treated or mock treated with 10 U *X. manihotis* α-1,2-fucosidase (New England Biolabs) at 37 °C for 16 hr. The reactions were stopped by heating at 100 °C for 10 min and desalted on mixed bed columns as described for the activity assay. After freeze-drying, samples were washed with water, re-dissolved in 20% 1-propanol, and analysed by HPTLC.

## Large-scale TbFUT1 assay and product purification

Acceptor (5 mM lacto-N-biose-β-O-methyl) and donor (2.5 mM GDP-Fuc) were incubated with 8 µg affinity-purified GST-TbFUT1 in 20 mM Tris-HCl, 25 mM KCl, pH 7.2 at 37 °C for 24 hr. The reaction products were desalted on a mixed-bed column and lyophilized (see above). The trisaccharide product was then isolated by normal phase liquid chromatography. The reactions were dissolved in 95% acetonitrile (ACN) and loaded on a Supelco SUPELCOSYL LC NH$_2$ HPLC column (7.5 cm × 3 mm, 3 µm) using an Agilent 1,120 Compact LC system. The purification was performed at 40 °C with a 0.3 ml/min flow rate. The column was equilibrated in 100% Buffer A (95% ACN, 5% H$_2$O) for 5 min, before applying a first gradient from 0% to 35% Buffer B (95% H$_2$O, 5% ACN, 15 mM ammonium acetate) over 25 min followed by a second gradient from 35% to 80% Buffer B over 20 min. The column was then re-equilibrated in 100% Buffer A for 10 min. About 1.5% of each fraction was potted onto a silica HPTLC plate and stained with orcinol/H$_2$SO$_4$ staining to identify the sugar-containing fractions. An aliquot from each orcinol-positive fraction was then analysed by HPTLC, and the ones containing the putative trisaccharide product were pooled and freeze-dried.

## Permethylation, ESI-MS analysis, and GC-MC methylation linkage analysis

Purified TbFUT1 reaction product was dried and permethylated by the sodium hydroxide method as described in *Ferguson, 1992*. Aliquots were used for ESI-MS and the remainder was subjected to acid hydrolysis followed by NaB$^2$H$_4$ reduction and acetylation (*Ferguson, 1992*). The resulting PMAAs were analysed using an HP6890 GC System equipped with an HP-5 column linked to a 5975C mass spectrometer (Agilent). For ESI-MS and ESI-MS/MS, the permethylated trisaccharide was dried and re-dissolved in 80% ACN containing 0.5 mM sodium acetate and loaded into gold-plated nano-tips (Waters) for direct infusion. The Q-Star XL mass spectrometer equipped with Analyst software (Applied Biosystems) was operated in positive ion mode using an ion spray voltage of 900 V and a collision voltage of 50–60 V in MS/MS mode.

## NMR

The purified TbFUT1 reaction product was exchanged in $^2$H$_2$O by freeze-drying and analysed by one-dimensional $^1$H-NMR and two-dimensional $^1$H-ROESY. All spectra were acquired on a Bruker Avance spectrometer operating at 500 MHz with a probe temperature of 293 K.

## Generation and expression of epitope-tagged TbFUT1 constructs

*TbFUT1* was introduced in two different sites of pLEW100HXM to yield HA$_3$-TbFUT1 and HA$_3$-TbFUT1-MYC$_3$. Starting from the pLEW100 vector, we inserted a 99 bp adapter oligo encoding for the following sequence *Hind*III/*Nde*I/*Asc*I/*Bbv*CI/*Xba*I/*Bst*BI/*Xho*I/<u>HA</u>/*Pac*I/*Bam*HI, where <u>HA</u> stands for a single HA tag (O1 and O2 in *Supplementary file 2*). The HA tag was then replaced with a MYC$_3$ tag amplified from a pMOTag43M plasmid, creating the following multiple cloning site: *Hind*III/*Nde*I/*Asc*I/*Bbv*CI/*Xba*I/*Bst*BI/*Xho*I/<u>MYC$_3$</u>/*Pac*I/*Bam*HI. This vector was given the name pLEW100XM. *TbFUT1* was amplified from a plasmid source using primers P15/P16 and inserted via *Hind*III/*Xho*I. pLEW100XM was then used to introduce an HA$_3$ adapter which would allow N-terminal tagging of a protein of choice. The adapter consisted of the O3/O4 primer pair (*Supplementary file 2*). The primer pair was fused and inserted into *Hind*III/*Asc*I restriction sites of pLEW100XM creating the following multiple cloning site: *Hind*III/**HA$_3$**/*Asc*I/*Bbv*CI/*Xba*I/*Bst*BI/*Xho*I/**MYC$_3$**/*Pac*I/*Bam*HI. This plasmid was termed pLEW100HXM. TbFUT1 was amplified using primers P21/P22 for HA$_3$-TbFUT1 and P23/P16 for HA$_3$-TbFUT1-MYC$_3$ (*Supplementary file 2*). The two plasmids were purified and electroporated into BSF cells as described below.

## Generation of *TbFUT1* conditional null mutants

About 500 bp of the 5′ and 3′ untranslated regions (UTRs) immediately flanking the *TbFUT1* ORF were amplified from *T. brucei* 427 genomic DNA (gDNA) using primers P3/P5 and P4/P6 (*Supplementary file 2*) and linked together by PCR, and the final product was ligated into pGEM-5Zf(+). Antibiotic resistance cassettes were cloned into the *Hind*III/*Bam*HI restriction sites between the two UTRs to generate constructs either containing puromycin acetyltransferase (*PAC*) or blasticidin S deaminase (*BSD*). To allow cloning of the antibiotic resistance cassettes using *Hind*III/*Bam*HI, a *Bam*HI site present in the 5′-UTR was first mutated to ggctcc using the QuickChange Site-Directed Mutagenesis

Kit (Stratagene) and primers P7/P8. These two constructs were used to replace the two gene copies in procyclic form parasites. In addition, a hygromycin (HYG)-based *TbFUT1* gene replacement cassette was generated with longer UTRs (1.25 kb) amplified using primers P9/P10 (5′ UTRs) and P11/P12 (3′UTRs). To avoid the endogenous 5′ UTR site, first the 3′UTR was introduced via *Bam*HI/*Sac*I then the 5′UTR via *Spe*I/*Hind*III. The HYG resistance cassette replaced both copies of *TbFUT1* in bloodstream form parasites. The tetracycline-inducible ectopic copy construct was generated by amplifying the *TbFUT1* ORF from *T. brucei* gDNA (primers P13/P14) and cloning the resulting PCR product into pLEW100. Additionally, pLEW100XM was generated which allowed universal tagging of a protein of interest with a C-terminal MYC$_3$ tag (see details below). Linearized DNA was used to transform the parasites as previously described (*Burkard et al., 2007*; *Güther et al., 2006*; *Wirtz et al., 1999*). The genotype of the transformed parasites was verified by Southern blot as detailed below. The TbFUT1-MYC$_3$ cell line was used as parental strain for the BSF *TbFUT1* cKO cell line by eliminating the MYC$_3$ tag and inserting a cassette comprised of the native (untagged) *TbFUT1* gene, a short intergenic region (IGR), and the PAC resistance gene via homologous recombination. A C-terminal fragment of the *TbFUT1* ORF (nucleotides 444 to end) was amplified with primers P28/29 (*Supplementary file 2*) and cloned into the Kpn*I*/Apa*I* restriction sites of pMOTag4YH-PAC (*Oberholzer et al., 2006*), a kind gift of Prof. Thomas Seebeck (University of Bern). The eYFP-3xHA cassette encoded in the plasmid was removed by Xho*I*/Sal*I* digestion followed by re-ligation. Finally, the rDNA homology sequence was digested with Not*I*/Sac*I* from pLEW100HXM and cloned into these same sites downstream of the puromycin resistance gene (PAC), resulting in pMOTag-*TbFUT1*-IGR-PAC-rDNA. The linearized plasmid was electroporated in *BSF TbFUT1-MYC$_3$* cKO. Homologous recombination via the *TbFUT1* C-terminus and rDNA sequences resulted in generation of BSF *TbFUT1* cKO (*Figure 5C*).

## Southern blotting

Genomic DNA was prepared from $5 \times 10^7$ or $1 \times 10^8$ cells using DNAzol (Helena Biosciences) and digested with 0.1 mg/ml RNAse I. The gDNA (5 µg/lane) was digested with the appropriate restriction endonucleases and probed with the ORFs of *TbFUT1*, puromycin acetyltransferase gene (*PAC*), hygromycin B phosphotransferase (*HYG*), or blastacidin S deamidase gene (*BSD*) generated using the PCR DIG Probe Synthesis Kit (Roche). The blot was developed according to the manufacturer's instructions.

## Northern blotting

Total RNA was prepared from $5 \times 10^6$ to $1 \times 10^7$ cells using the RNeasy MIDI Kit (Qiagen) according to the manufacturer's instructions. The RNA was separated on a 2% agarose-formaldehyde gel, blotted, and detected using the Northern Starter Kit (Roche). Probes were designed based on the DIG RNA Labelling Kit T7 (Roche), and *TbFUT1* and alpha-tubulin (Tb427.01.2340) templates were amplified from *T. brucei* bloodstream form gDNA using primers P17/P18 and P19/P20 (*Supplementary file 2*), respectively. Total RNA and DIG labelled probes were quality checked by capillary electrophoresis on an Agilent BioAnalyzer 2100.

## Growth curves

Growth of parasites lacking TbFUT1 was analysed by determining cell counts for three clones each of PCF *TbFUT1* cKO and BSF *TbFUT1* cKO with either tagged of untagged TbFUT1. All cell lines were washed in media without tetracycline (tet) and inoculated at a concentration of $5 \times 10^5$ parasites/ml (procyclic form) and $2 \times 10^4$ parasites/ml (bloodstream form). Parasites were grown plus and minus tetracycline and in biological duplicates (PCF) or triplicates (BSF). Cells were counted every 24 hr using a CASY Cell Counter + Analyser system. To keep parasites in log phase, cultures were diluted back to the initial inoculum concentration every 48 hr.

### Preparation of anti-FUT1 antibody

The *Nde*I site within the *TbFUT1* sequence was eliminated using the QuickChange Lightning Site-Directed Mutagenesis Kit (Agilent) and primers P24/P25. The resulting mutagenized ORF was then amplified with primers P26/P27 and cloned into pET15b, resulting in an N-terminally truncated construct encoding Δ$_{32}$TbFUT1 fused to an N-terminal hexahistidine tag (HIS$_6$) with PreScission plus (PP) protease cleavage site. *E. coli* BL21 gold (DE3) cells were allowed to express His$_6$-PP-TbFUT1

overnight at 25 °C in auto-inducing media (5052-NPS-MgSO₄). Cells were harvested by centrifugation and incubated for on ice in 50 mM Tris-HCl pH 7.3, 0.15 M NaCl, 1 mM DTT (Buffer A) with EDTA-free Complete Protease Inhibitors Tablet (Roche), before being disrupted in a French Press. Since all of the protein was insoluble within inclusion bodies, a refolding procedure was applied. The cell lysate was spun down for 20 min at 20,000 × g and the pellet was washed in ice-cold PBS with EDTA-free Complete Protease Inhibitors Tablet (Roche). Cell debris layering the inclusion body pellet was scratched off and the pellet washed again. This process was repeated three times. The inclusion body pellet was washed with PBS + 0.5% TX-100 and sonicated on ice. After digestion of residual DNA and RNA, the pellet was washed with PBS + 0.1% Tween-20, sonicated, and washed in TBS. The washed inclusion bodies were resuspended in 8 M urea/TBS and incubated overnight at 37 °C, 200 rpm. Insoluble material was spun down and the supernatant slowly diluted down under agitation to 2 M urea with TBS at room temperature. After centrifugation, the supernatant concentrated to 2 mg/ml in a 30 kDa cut-off VivaSpin concentrator. Re-solubilized protein (2 mg) was sent to DC Biosciences Ltd for production of polyclonal rabbit antiserum. The IgG fraction from the hyperimmunized rabbit serum was purified on 2 × 1 ml HiTrap Protein G HP columns (GE Healthcare) and then affinity-purified on GST-TbFUT1 fusion protein coupled to a HiTrap NHS-Activated HP column (GE Healthcare). Finally, the IgG fraction was adsorbed against PCF *TbFUT1* cKO cell lysates prepared on day 9 without tetracycline, a time point at which they are virtually free of TbFUT1. The cell lysate was prepared by sonication of 2 × 10⁸ cells on ice in 100 mM sodium phosphate buffer pH 7 including EDTA-free Complete Protease Inhibitors (Roche) and 0.01% TX-100. After addition of purified TbFUT1-specific IgGs to the cell lysate, the solution was incubated rotating overnight at 4 °C, concentrated in a VivaSpin 30 kD cut-off filter, and stored in 50 % glycerol at –20 °C. Final estimated rabbit anti-TbFUT1 IgG concentration was 50 µg/ml.

## Immunofluorescence microscopy

Late log phase *T. brucei* bloodstream form or procyclic cells were fixed in 4% PFA/PBS in solution at a concentration of 5 × 10⁶ cells/ml. When using MitoTracker Red CMX Ros, cell cultures were spiked with a 25 nM concentration over 20 min before harvesting. Fixed parasites were permeabilized in 0.5% TX-100 in PBS for 20 min at room temperature before blocking in 5 % fish skin gelatine (FSG) containing 0.1% TX-100 and 10% normal goat serum. Alternatively, procyclic cells were permeabilized in 0.1% TX-100 in PBS for 10 min and blocked in 5% FSG containing 0.05% TX-100 and 10% normal goat serum. Primary antibody incubations were performed in 1% FSG, 0.1% TX-100 in PBS. The following primary antibodies were used in 1:1000 dilutions: rabbit anti-TbFUT1 IgG (see above), rabbit anti-GRASP IgG (gifts from Prof. Graham Warren, University College London, and Prof. Chris de Graffenreid, Brown University), rabbit anti-ATPase IgG (gift from Prof. David Horn, University of Dundee), mouse anti-HA IgM (Sigma), rabbit anti-HA IgG (Invitrogen), and mouse anti-MYC IgM (Millipore). Anti-FAZ was used at 1:2 and anti-PFR at 1:10 (both antibodies were kind gifts of Prof. Keith Gull, University of Oxford). After washing, the coverslips were incubated with a 1:500 dilution of respective secondary Alexa Fluor goat IgG, like anti-mouse 488 and 594, as well as anti-rabbit 488 and 594 (Life Technologies). Coverslips were mounted on glass slides with Prolong Gold DAPI antifade reagent (Invitrogen). Microscopy was performed on a DeltaVision Spectris microscope (GE Healthcare), and images were processed using Softworx. For *Figure 6—figure supplement 3*, microscopy was performed on a Zeiss LSM 700 confocal microscope. All images were further processed using Fiji (*Schindelin et al., 2012*).

## Scanning electron microscopy

Samples were prepared and analysed as described in *Turnock et al., 2007*. Briefly, PCF *TbFUT1* cKO cells were grown in the presence or absence of tetracycline and fixed on days 6 and 12 in SDM-79 with 2.5% glutaraldehyde, and the samples were further prepared by the Dundee Imaging Facility, School of Life Sciences. Cells were collected on 1 µM Shandon Nuclepore membrane filters, rinsed twice in 0.1 M PIPES buffer pH 7.2, and treated with 0.2% osmium tetroxide overnight. Samples were then washed in water and in a gradient of ethanol solutions to dehydrate them. After critical point drying on a BALTec critical point dryer D30, the samples were mounted on aluminium stubs that were then coated with 40 nM gold/palladium. The stubs were examined in a Philips XL30 environmental scanning electron microscope operating at an accelerating voltage of 15 kV.

## Cell volume determination

The average cell volume was obtained by analysing PCF *TbFUT1* cKO and *TbGMD* cKO on a CASY Cell Counter + Analyser system. Procyclic form cultures grown in the presence or absence of tetracycline were diluted 1:10 in 1× PBS before being diluted in the CASY-ton solution for measurements (>4000 cells/measurement). Two biological replicates were analysed for procyclic form grown in the absence of tetracycline. The p values were calculated performing a two-sided *t*-test in RStudio.

## Sedimentation assay

The assay was adapted from *Bastin et al., 1999*. Briefly, about $5 \times 10^6$ *T. brucei* PCF *TbFUT1* cKO or *TbGMD* cKO cells were resuspended in 1 ml of SDM-79 medium and transferred into a 2 ml polystyrene cuvette (Sarstedt). The cuvettes were incubated at 28 °C (CPS-controller, Shimadzu) in a UV-1601 Spectrophotometer (Shimadzu) and the optical density at 600 nm measured every 30 min. SDM-79 medium was used as a blank. Two biological replicates were analysed for each cell line grown in the presence or absence of tetracycline.

## *L. major* complementation by 'plasmid shuffling'

For the plasmid swap experiments described in *Figure 10*, the *TbFUT1* ORF was amplified by PCR and inserted into the expression vector pIR1NEO yielding pIR1NEO-TbFUT1 (B7148). *L. major* parasites were grown as promastigotes in M199 medium, transfected, and plated on selective media for clonal lines as described previously (*Guo et al., 2017*). 'Plasmid shuffling' to swap an episomal *LmjFUT1*-expressing construct with the ones expressing-TbFUT1 (±HA tag) was performed as described in the text and *Figure 10A*, following procedures described in *Guo et al., 2017*.

## Acknowledgements

We thank Gina MacKay and Art Crossman (University of Dundee) for performing the NMR experiment and helping with the data analysis. We would also like to thank Alan R Prescott (Division of Cell Signalling and Immunology, University of Dundee) for his generous help with the confocal microscopy and Martin Kierans for preparing the samples for scanning electron microscopy. We are also grateful to Keith Gull (University of Oxford), Graham Warren (University College London), Chris de Graffenreid (Brown University), Thomas Seebeck (University of Bern), and Daan van Aalten and David Horn (University of Dundee) for providing reagents. This work was supported by a Wellcome Trust Investigator Award (101842) to MAJF, University of Dundee/BBSRC PhD studentship to GB, NIH Grant R01-AI31078 to SMB, and postdoctoral fellowship to HG.

## Additional information

### Funding

| Funder | Grant reference number | Author |
|---|---|---|
| Wellcome Trust | 101842 | Michael A J Ferguson |
| National Institute of Allergy and Infectious Diseases | R01-AI31078 | Stephen Beverley |

The funders had no role in study design, data collection and interpretation, or the decision to submit the work for publication.

### Author contributions

Giulia Bandini, Conceptualization, Formal analysis, Investigation, Methodology, Validation, Visualization, Writing – original draft, Writing – review and editing; Sebastian Damerow, Conceptualization, Formal analysis, Investigation, Methodology, Validation, Visualization, Writing – original draft; Maria Lucia Sempaio Guther, Formal analysis, Investigation, Methodology, Supervision; Hongjie Guo, Formal analysis, Investigation, Methodology, Visualization; Angela Mehlert, Formal analysis, Methodology, Visualization; Jose Carlos Paredes Franco, Formal analysis, Investigation, Methodology; Stephen Beverley, Conceptualization, Formal analysis, Funding acquisition, Supervision, Visualization,

Writing – original draft, Writing – review and editing; Michael AJ Ferguson, Conceptualization, Formal analysis, Funding acquisition, Methodology, Project administration, Resources, Supervision, Visualization, Writing – original draft, Writing – review and editing

### Author ORCIDs
Giulia Bandini http://orcid.org/0000-0002-8885-3643
Michael AJ Ferguson http://orcid.org/0000-0003-1321-8714

### Decision letter and Author response
Decision letter https://doi.org/10.7554/eLife.70272.sa1
Author response https://doi.org/10.7554/eLife.70272.sa2

## Additional files

### Supplementary files
• Supplementary file 1. Summary of BLASTp searches for putative fucosyltransferases in the three kinetoplastids genomes.

• Supplementary file 2. Primers and oligonucleotides used in this study.

• Source data 1. This folder includes the raw data for the following figures: *Figure 2*, *Figure 2—figure supplement 1*, *Figure 2—figure supplement 2*, *Figure 3—figure supplement 1*, and *Figure 3—figure supplement 2*.

• Source data 2. This folder includes the raw data for *Figure 5—figure supplement 1*.

• Source data 3. This folder includes the raw data for *Figure 6D–F*.

• Transparent reporting form

### Data availability
All data is available in the manuscript. No sequencing, proteomics or protein structural data were generated.

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
