## [Decision Letter]

**Acceptance summary:**

This study describes an unusual fucosyltransferase in the parasitic protist, *Trypanosoma brucei*, that is essential for growth. Unlike other eukaryotic fucosyltransferases, the *T. brucei* enzyme is localized to the mitochondrion, rather than the secretory pathway or cytoplasm. The trypanosome enzyme belongs to the CAZy GT11 family and transfers fucose to synthetic glycan substrates, although the in vivo substrate and function of this enzyme remains enigmatic. It is essential for other typanosomatid parasites making it a potential drug target.

**Decision letter after peer review:**

Thank you for submitting your article "An essential, kinetoplastid-specific GDP-Fuc: β-D-Gal α-1,2-fucosyltransferase is located in the mitochondrion of *Trypanosoma brucei*" for consideration by *eLife*. Your article has been reviewed by 3 peer reviewers, and the evaluation has been overseen by a Reviewing Editor and Dominique Soldati-Favre as the Senior Editor. The following individual involved in review of your submission has agreed to reveal their identity: Álvaro Acosta Serrano (Reviewer #2).

Essential revisions:

1) While the data supporting a mitochondrial localization for TbFUT1 is strong, direct evidence for matrix localization (as implied in the discussion), versus a localization in the outer membrane or inter-membrane space is missing. Inclusion of additional data supporting the matrix location of TbFUT1 (including data from previous proteomic analyses of *T. brucei* mitochondrion, protease protection experiments etc) is recommended and/or the conclusions in the discussion suitably tempered.

2) Given that TbFUT1 exhibits a high level of non-productive hydrolysis of GDP-Fuc in the in vitro assay and the substrate of TbFUT1 in vivo is unknown, it is strongly recommended that the authors confirm that TbFUT1 does not have promiscuous sugar nucleotide activity and utilizes GDP-Man as well as GDP-Fuc.

3) Additional data demonstrating that the N-terminal sequence in TbFUT1 functions as a mitochondrial targeting signal – here or in follow up studies – would also enhance the conclusions.

*Reviewer #1:*

In 2007 the authors published a paper demonstrating that GDP-fucose, the donor substrate for fucosyltransferases (FUTs), is essential for *Trypanosoma brucei* growth in procyclic and bloodstream forms (Turnock et al., 2007). At that time, no FUTs or fucosylated proteins were known to exist in *T. brucei*. Here, using bioinformatic analysis, the authors identified a single FUT homologue in the *T. brucei* genome, TbFUT1, which belongs to the GT11 CAZy family of a1,2-FUTs. They expressed TbFUT1 in *E. coli* and demonstrated that it is a FUT that transfers fucose to acceptor substrates, preferentially with a terminal Galb1,3GlcNAc. Glycosidase digestions, mass spectral and NMR analyses demonstrated addition of fucose to the 2'-OH of the galactose in α-linkage. They generated conditional null (with a tetracycline-inducible promoter) alleles of TbFUT1 in both bloodstream (BSF) and procyclic (PCF) forms of the parasite. Deletion of TbFUT1 resulted in a significant growth defect in both forms when grown in vitro. Surprisingly, TbFUT1 localized to the single mitochondrion in *T. brucei*, and deletion of TbFUT1 disrupted mitochondrial activity. A homolog exists in *Leishmania major* (LmjFUT1), which has recently been reported to be essential for *L. major* growth (Guo et al. 2021). The authors showed that TbFUT1 rescued the growth defect in LmjFUT1 mutants, and also localized to the mitochondrion in *L. major*.

This study is the first to identify a FUT in *T. brucei*, and the first to identify a FUT localized to mitochondria in any species. FUTs are typically localized in the Golgi, ER, or as more recently described, in the cytoplasm, but this is the first demonstration of a FUT in mitochondria. Only one glycosyltransferase has been reported in mitochondria, O-GlcNAc transferase (OGT). OGT transfers GlcNAc directly to protein, so no additional glycosylation machinery is necessary. The fact that TbFUT1 prefers a Galb1,3GlcNAc acceptor substrate strongly suggests additional glycosyltransferases will also be localized to the mitochondria in *T. brucei* and *L. major*. Thus, these are highly significant and novel studies. They are well done and rigorously performed. A few minor suggestions for clarity and presentation of the data are needed.

*Reviewer #2:*

The authors used a nice combination of biochemical methodologies, including NMR and mass spectrometry, to work out the substrate specificity of this activity; they concluded that it mainly transfers alpha1-2Fuc to Galbeta1-3GlcNAc but not Galbeta1-4GlcNAc dissacharides. Interestingly, Galbeta1-3GlcNAc residues are highly abundant on surface glycoconjugates expressed by the procyclic (midgut) stage and also as terminal sugars on N-glycans from flagellar pocket glycoproteins expressed in the blood stages, but their mitochondrial presence has never been found in any organism. It remains to be determined what is the natural substrate of this enzyme and whether fucosylation indeed occurs in the mitochondrion of kinetoplastid organisms.

I only have few comments and no further experiments are requested.

1. Did the authors try to determine the exact location of TbFUT1 within the parasite mitochondrion, for instance using TEM immunogold? This could help understanding access to GDP-Fuc (although I believe the enzyme is facing the cytosol) and also the possible location of the natural acceptor molecule(s).

2. In relation with the previous point, given the challenges in trying to localise fucosylated glycans using fucose-specific lectins, I wonder if there is a precedent for detecting terminal β-Gal residues on the trypanosome mitochondrion using lectins.

3. I found interesting that N-terminal tagging of TbTUF1 sends the protein to the Golgi apparatus. This seems like a great coincidence for a protein that normally would be predicted to be Golgi-resident, so I wonder if there is any identifiable Golgi targeting sequence within TbTUF1. Also, there was any attempt to localise the protein after deletion of the mitochondrial signal (no tagging)?

*Reviewer #3:*

A great strength of the manuscript is the meticulous, very convincing biochemical analysis of the recombinant trypanosomal fucosyltransferase. No surprise since it has been done by leaders in the field. Also the immunofluoresecence localization is convincing, within the limits of the technique. The authors went through the great effort of producing a monospecific antiserum against the fucosyltransferase which allows them to localize the natively expressed untagged protein. The conditional knock out cell lines for both cell cycle stages unambiguously demonstrate that the enzyme is essential for cell growth.

The manuscript is well written, has clearly composed figures and the results are presented in a logical way. I agree with the authors general conclusions. My major criticism is that I think both the localization of the enzyme as well as its physiological properties could have been analyzed in some more detail using the great tools the authors produced in their study (cell lines, antiserum). Having said that I am aware that characterizing the exact function of the protein might be beyond the scope of this study.

TbFUT1 appears to have a convincing motif IV and as far as I understood the biochemical assays have been done with the full length TbFUT1 fused to GST which therefore also contains motif IV. In the paper it is argued that motif IV might be part of the putative presequence that would be removed after import. My question therefore is whether TbFUT1 indeed has a N-terminal mitochondrial targeting signal and if yes whether removal of this sequence interferes with the enzymatic activity, because it would truncate the motif IV. If a N-terminally truncated C-terminally tagged TbFUT1 does not localize to mitochondria anymore this would confirm the presence of a presequence. Would it be possible to compare the size of the imported TbFUT1 with the full length protein, to get evidence for proteolytic processing of the protein? Instead it could be shown that a truncated recombinant protein is still functional etc.

The IF localization is convincing, however it does not exclude that the proteins localizes to the surface of the mitochondrion only. It would help to show that the protein is really localized inside mitochondria. Biochemical analysis could be useful here, if the proteins is inside mitochondria it should be resistant to added proteases in isolated organelles. (If it has a bona fide presequence that is cleaved, this would indicate that its import into the matrix, see above). Have the authors checked whether the protein was detected in previously published mitochondrial proteomes, this could provide additional evidence for its mitochondrial localization.

A proteomic analysis of uninduced and induced conditional knock out cell line could provide valuable information why TbFUT1 is essential for growth and provide hints to its function. The same is true for pull down assays to identify its interaction partners.

It would be helpful to precisely time the observed phenotypes in the conditional knock out cell lines, to know whether loss of the membrane potential and increase in cell volume are early phenotypes and to determine whether they have a defined order (see below).

The authors show that ablation of TbFUT1 causes an increase in cell volume. Could this be the "Nozzle phenotype" that has been described before by the Matthews group and others.

The loss of membrane potential is observed in bloodstream form cells 5 days after removal of Tet. This is two days into the growth phenotype and therefore might be a secondary effect. The reduction of the ATPase suggests the same thing. Please discuss.

---

## [Author Response]

Essential revisions:1) While the data supporting a mitochondrial localization for TbFUT1 is strong, direct evidence for matrix localization (as implied in the discussion), versus a localization in the outer membrane or inter-membrane space is missing. Inclusion of additional data supporting the matrix location of TbFUT1 (including data from previous proteomic analyses of *T. brucei* mitochondrion, protease protection experiments etc) is recommended and/or the conclusions in the discussion suitably tempered.

Thank you for these comments. We agree that the localisation claims need to be tempered. Additional literature (proteomic data) information has also been included (see also response to Reviewer #2 and #3) and the discussion has been edited to temper the conclusions on the TbFUT1 localisation.

Lines 465-483: “Immunofluorescence analysis, using an affinity-purified antibody raised against native TbFUT1 as well as C-terminal epitope tagging, suggest that TbFUT1 is located in the parasite mitochondrion. […] These studies will also assess the fate of motif IV and its function in TbFUT1 enzymology.”

Lines 495-502: “Firstly, for a TbFUT1 that localises to either the intermembrane space or the mitochondrial matrix to be active, GDP-Fuc would need to be imported into the mitochondrion, which would require the presence of an uncharacterized mitochondrial GDP-sugar transporter. Secondly, TbFUT1 appears to be an α-1,2-FUT that decorates glycans terminating in Galβ1,3GlcNAc, suggesting either that additional uncharacterized glycosyltransferases and nucleotide sugar transporters may be present in the parasite mitochondrion, or that a glycoconjugate substrate may be assembled in the cytoplasm or secretory pathway and then somehow translocated to or imported into the mitochondrion to be modified by TbFUT1.”

Lines 510-513: “In conclusion, TbFUT1 is an essential Α-1,2-FUT that localises to either outer membrane or inner compartments of the parasite mitochondrion and presents orthologues throughout the kinetoplastida. As described in (Guo et al. 2021), both activity and mitochondrial localization of the *L. major* homologue are required for parasite viability,”

2) Given that TbFUT1 exhibits a high level of non-productive hydrolysis of GDP-Fuc in the in vitro assay and the substrate of TbFUT1 in vivo is unknown, it is strongly recommended that the authors confirm that TbFUT1 does not have promiscuous sugar nucleotide activity and utilizes GDP-Man as well as GDP-Fuc.

Thank you for this helpful comment. We have performed additional work and utilized the GDP-Glo assay to test sugar nucleotide specificity and now report that the enzyme will turnover GDP-Fuc, as expected, but not GDP-Man (or GDP-Glc). This information now appears in lines 204-211 and the assay methodology has been added to the Methods section (lines 561-572). We have also added the person who performed these assays as a co-author (J.C. Paredes Franco).

Results (Lines 204-211): “Finally, given the propensity for TbFUT1 to transfer [3H]Fuc from GDP-[3H]Fuc to water, producing free [3H]Fuc and presumably GDP, we set up a GDP-GloTM assay similar to that used for LmjFUT1 (Guo et al., 2021) to monitor the turnover by recombinant TbFUT1 of non-radioactive GDP-Fuc to GDP (see Methods). In this assay we could see TbFUT1-dependent turnover of GDP-Fuc to GDP in the absence of acceptor substrate (70±2 pmol) and the stimulation of turnover in the presence of LNB acceptor substrate (265±13 pmol), consistent with the results in the radiometric assay. Under the same conditions there was no detectible turnover of either GDP-Man or GDP-Glc, showing that TbFUT1 is specific, or at least highly-selective, for GPD-Fuc as donor substrate.”

Methods (lines 561-572): “Luminescence assay. Recombinant GST-TbFUT1 fusion protein was expressed in *E. coli* as described above, except that cell lysis was performed without lysozyme using a Cell Disruptor (Constant Systems) in place of a French press. […] A calibration curve using 25 μl aliquots of GDP from 0 to 25 μM in assay buffer were developed in parallel. All measurements were performed in triplicate.”

3) Aadditional data demonstrating that the N-terminal sequence in TbFUT1 functions as a mitochondrial targeting signal – here or in follow up studies – would also enhance the conclusions.

Further studies will be performed to study the function of the N-terminal sequence in both mitochondrial targeting and enzyme activity, as detailed in the reply to the reviewer 3 first comment. In the meantime, we have updated the discussion as indicated above (Lines 465-483).

Reviewer #2:I only have few comments and no further experiments are requested.1. Did the authors try to determine the exact location of TbFUT1 within the parasite mitochondrion, for instance using TEM immunogold? This could help understanding access to GDP-Fuc (although I believe the enzyme is facing the cytosol) and also the possible location of the natural acceptor molecule(s).

No, we did not. Cryo-immunoEM performed by Guo et al., suggests that the LmjFUT1 localises to the mitochondrial lumen. The assumption would be that TbFUT1 also localises to this mitochondrial compartment, but this indeed needs to be experimentally confirmed. A localisation in the mitochondrial lumen or the inter-membrane space would not suggest easy access to cytosolic GDP-Fuc pools and the need for transporters, since *T. brucei* sugar nucleotide biosynthetic enzymes are predominantly localised in the glycosomes. Consequently, these sugar precursors would need to be transported out of these organelles to the cytosol and then into ER, Golgi and, as we suggest here, mitochondrion.

2. In relation with the previous point, given the challenges in trying to localise fucosylated glycans using fucose-specific lectins, I wonder if there is a precedent for detecting terminal β-Gal residues on the trypanosome mitochondrion using lectins.

Thank you for the suggestion. This is definitely an experiment worth trying in the search for TbFUT1 endogenous substrates. We concentrated our efforts on the fucose-specific lectins, although unsuccessfully, and did not look at staining of either wild type or TbFUT1-deficient cells with β-Gal specific lectins. Previously published IFA with the β-Gal specific lectin RCA120 labelled the flagellar pocket of bloodstream form trypanosomes, where the N-glycans carrying poly-LacNAc repeats are abundant. However, no permeabilisation step was performed and so any organelle labelling would have been missed (Atrih et al., 2005, JBC, 280:865-871)

3. I found interesting that N-terminal tagging of TbTUF1 sends the protein to the Golgi apparatus. This seems like a great coincidence for a protein that normally would be predicted to be Golgi-resident, so I wonder if there is any identifiable Golgi targeting sequence within TbTUF1. Also, there was any attempt to localise the protein after deletion of the mitochondrial signal (no tagging)?

We agree with the reviewer that the localization of the N-terminally tagged TbFUT1 to the Golgi is an odd coincidence. It is worth noting that *L. major* FUT1 tagged at the N-terminus localizes to the cytosol as described in Guo et al., 2021. The same is observed of a tagged LmjFUT1 lacking the N-terminal mitochondrial targeting sequence. Furthermore, data from the TrypTag project suggests N-terminal tagging of TbFUT1 in PCF results in cytosolic and nucleoplasm localisation (tryptag.org, Dean et al., 2017, Trends Parasitol, 33:80-82).

As briefly mentioned in the Results section, the same algorithms that suggested untagged TbFUT1 was likely to localize to the mitochondrion, confirm this prediction for the C-terminally MYC3-tagged protein, but not the N-terminal HA3-tagged versions. However, the predictions for either HA3-TbFUT1 or HA3-TbFUT1-MYC3 do not support the observed Golgi localisation and are in better agreement with the cytosolic staining observed for the comparable LmjFUT1 construct or PCF expression.

In the case of mammalian and yeast Golgi-resident glycosyltransferases (GTs), retention of these type II membrane proteins to the Golgi seems to be dependent on a combination of features in the cytosolic tail, TM and stem domains however no specific sequence or motif has been identified so far (Tu and Banfield 2010, Cell Mol Life Sci, 67:29-41). There is no predicted signal peptide for TbFUT1 and the prediction of a type II topology is weak (see Reviewer 1, point 3) and thus it does not fit with what is known about retention in the Golgi.

We did not try to localize TbFUT1 lacking the putative mitochondrial targeting sequence, but the experiment is very much worth performing to better understand this fucosyltransferase behaviour (see also Reviewer 3, point 1).

Reviewer #3:TbFUT1 appears to have a convincing motif IV and as far as I understood the biochemical assays have been done with the full length TbFUT1 fused to GST which therefore also contains motif IV. In the paper it is argued that motif IV might be part of the putative presequence that would be removed after import. My question therefore is whether TbFUT1 indeed has a N-terminal mitochondrial targeting signal and if yes whether removal of this sequence interferes with the enzymatic activity, because it would truncate the motif IV. If a N-terminally truncated C-terminally tagged TbFUT1 does not localize to mitochondria anymore this would confirm the presence of a presequence. Would it be possible to compare the size of the imported TbFUT1 with the full length protein, to get evidence for proteolytic processing of the protein? Instead it could be shown that a truncated recombinant protein is still functional etc.

Thank you for your comments. We agree that the presence and role of the predicted presequence should be further investigated as well as the effect of cleavage of motif IV if this does occur, as would be predicted, upon import in the mitochondrial matrix. This of course would not be the case if TbFUT1 localises in the intermembrane space or indeed the outer membrane. Therefore, these experiments are closely connected to the future work to confirm the mitochondrial localisation and determine where in the mitochondrion TbFUT1 is found, i.e., matrix, intermembrane space or outer membrane (discussed in the point below, in the response to the essential revisions and also in the first comment by Reviewer #2). It is worth noting that Guo et al., have shown that LmjFUT1 presequence is required for mitochondrial localisation and rescue of the knockout parasites. In our manuscript we show that TbFUT1 can also rescue *L. major* TbFUT1-deficient promastigotes and localises to the Leishmania mitochondrion, suggesting that TbFUT1 should also be imported in the mitochondrion. Nevertheless, we agree that evidence of presequence requirement needs to be determined in *T. brucei* parasites to confirm this model.

In this revised manuscript we have followed the editorial suggestion and edited the discussion to tamper the conclusion on the mitochondrial localisation. We have also added a comment on the specific issue of motif IV (Lines 478-483).

The IF localization is convincing, however it does not exclude that the proteins localizes to the surface of the mitochondrion only. It would help to show that the protein is really localized inside mitochondria. Biochemical analysis could be useful here, if the proteins is inside mitochondria it should be resistant to added proteases in isolated organelles. (If it has a bona fide presequence that is cleaved, this would indicate that its import into the matrix, see above). Have the authors checked whether the protein was detected in previously published mitochondrial proteomes, this could provide additional evidence for its mitochondrial localization.

TbFUT1 was identified in the mitochondrial proteome of procyclic form *T. brucei*, albeit with only one spectral count from one unique peptide (Panigrahi et al., 2009). Although this is not a high confidence assignment, it is worth noting this is the only proteomic dataset deposited on TriTrypDB that identified TbFUT1. TbFUT1 protein turnover could not be determined in a quantitative analysis performed in our lab (Tinti et al., 2019), suggesting it might be a low abundance protein. We have added these comments to the discussion (Lines 467-472).

Further studies to define the localization of TbFUT1 in the mitochondrion are indeed required and will include analysis of the presequence and assessment of proteolytic cleavage, as discussed above. We thank the reviewer for the suggestion of the protease resistance assay, this would be an elegant biochemical assay to define if TbFUT1 is found on the surface or inside the parasite mitochondrion.

A proteomic analysis of uninduced and induced conditional knock out cell line could provide valuable information why TbFUT1 is essential for growth and provide hints to its function. The same is true for pull down assays to identify its interaction partners.

We thank the reviewer for the suggestions. We agree that both experiments would be helpful to address the essential function of TbFUT1 and they will likely be part of future work to define the molecular basis of the phenotype. The identification of interaction partners will be particularly important to understand the molecular mechanism leading to the observed phenotype.

It would be helpful to precisely time the observed phenotypes in the conditional knock out cell lines, to know whether loss of the membrane potential and increase in cell volume are early phenotypes and to determine whether they have a defined order (see below).

We completely agree, see discussion below for the studies currently in progress to determine the phenotype of TbFUT1-deficient BSF and PCF. We feel that these studies are better suited to a follow up publication fully dedicated to the phenotype characterization.

The authors show that ablation of TbFUT1 causes an increase in cell volume. Could this be the "Nozzle phenotype" that has been described before by the Matthews group and others.

The IFA analysis showed in Figure 6 —figure supplement 1 (old Figure S7) suggests that in both TbFUT1 and TbGMD cells grown in non-permissive conditions the kinetoplast and the flagellar pocket might be further away from the posterior end compared to parasite grown in permissive conditions. The SEM shown in Figure 6 —figure supplement 2 does support this observation, which would be consistent with the elongated posterior end observed in the "nozzle phenotype". Staining with the YL1/2 antibody against tyrosinated tubulin would be required to confirm this is indeed the case.

The ongoing work on phenotype characterisation will include a more thorough analysis to determine if what is observed in TbFUT1-deficient PCF is indeed a ‘nozzle phenotype’ or just an indication of stressed cell, including (1) analysis of kinetoplast and nuclear DNA content during the cell cycle in conditional knockout cells grown in permissive and non-permissive conditions, (2) staining with YL1/2 antibody + DAPI.

The loss of membrane potential is observed in bloodstream form cells 5 days after removal of Tet. This is two days into the growth phenotype and therefore might be a secondary effect. The reduction of the ATPase suggests the same thing. Please discuss.

The growth phenotype becomes apparent between day 3 and day 4 after tetracycline removal as there is some variability between the cell lines expressing the tagged or untagged TbFUT1 (Figure 6B vs 6C). Since the analysis in Figure 9 was performed on the BSF TbFUT1 cKO carrying the untagged enzyme, both loss of membrane potential and reduction in ATPase staining were observed 24 h after the onset of the growth phenotype. Nevertheless, we agree with the reviewer that it will be important to determine if these mitochondrial-related phenotypes can be observed before the onset of the growth phenotype, as would be the expectation in the case of a direct effect on mitochondrial functionality. These experiments are being currently performed as part of the work to characterise the phenotype of PCF and BSF deficient parasites. For information, preliminary analyses suggest that assembly of complex V is defective in the BSF conditional KO grown in non-permissive conditions, supporting a direct role of TbFUT1 in mitochondrial function.